# Learning from Less: Guiding Deep Reinforcement Learning with Differentiable Symbolic Planning

## Abstract

Humans solve complex tasks by decomposing them into subtasks and flexibly revising plans based on observations. For example, when making coffee at a friend's place, you may initially plan to fetch coffee beans but skip this step upon noticing the machine is already full. In contrast, deep reinforcement learning agents lack such priors and typically require more interactions to achieve such adaptive behavior. This raises a key question: *How can we endow reinforcement learning (RL) agents with similar "human" priors, allowing the agent to learn with fewer training interactions?* To address this challenge, We propose **d**ifferentiable **sy**mbolic p**lan**ner (DYLAN), a novel framework that integrates symbolic planning into reinforcement learning. DYLAN functions as a differentiable reward model that incorporates human priors to shape rewards over intermediate subtasks, enabling more efficient exploration. Beyond reward shaping, DYLAN can also act as a high-level planner that composes logic-based options to generate new behaviors while avoiding common symbolic planning pitfalls such as infinite execution loops. We validate DYLAN on challenging exploration and generalization tasks, where it effectively overcomes reward sparsity through structured, symbolic guidance, and enables zero-shot generalization to novel tasks.

## 1. Introduction

The challenge of sparse rewards remains a significant barrier to the broader applicability and efficiency of RL methods (Ng et al., 1999; Andrychowicz et al., 2017; Ecoffet et al., 2021). In sparse environments, the reward signals are infrequent or delayed, making effective exploration difficult,

[1]Anonymous Institution, Anonymous City, Anonymous Region, Anonymous Country. Correspondence to: Anonymous Author <anon.email@domain.com>.

Preliminary work. Under review by the International Conference on Machine Learning (ICML). Do not distribute.

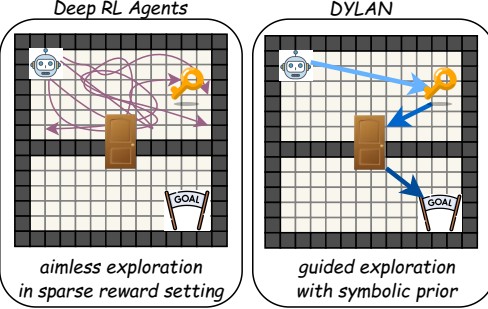

*Figure 1.* **Overcoming Exploration Bottlenecks with Human-Like Priors.** Deep RL agents (left) often explore aimlessly and inefficiently in sparse-reward environments. Dylan (right) bridges this gap by mimicking the human process of task decomposition. By sequencing logical subgoals (e.g.,"get key") through a differentiable symbolic planner, Dylan provides the guidance that eliminates wasteful exploration and accelerates the policy learning.

and the learning process can become prohibitively expensive in terms of computation (Pathak et al., 2017). Furthermore, sparse rewards often lack the granularity needed to explicitly guide agents toward desirable behaviors, frequently resulting in suboptimal or unintended outcomes as shown in Fig. 1. For example, while LLMs such as DeepSeek-R1 demonstrate a certain level of reasoning ability in solving complex problems, they may still exhibit issues such as language mixing or inconsistent linguistic patterns during multi-step reasoning—largely due to the limited structure provided by sparse rewards during post-training (Guo et al., 2025). Similarly, in robotics, sparse reward signals can lead agents to exploit unintended shortcuts or converge on behaviors that diverge from human expectations (Christiano et al., 2017a).

Prior works (Jaderberg et al., 2017; Ng et al., 1999) have explored reward shaping as additional learning signals to address this challenge. These signals are typically derived from potential-based functions (Ng et al., 1999), expert-crafted heuristics (Kober et al., 2013; Devlin et al., 2011), or learned reward models based on human preferences (Christiano et al., 2017b) and auxiliary tasks (Jaderberg et al., 2017). While these approaches can reduce the amount of data required for learning to some extent, a fundamental question remains: *Can we design a reward model that not*

*only reduces RL agent's training interactions, but also remains interpretable and aligns with similar 'human' intent?*

To address this question, we propose **Dylan**, a differentiable symbolic planner that serves as a reward model to guide reinforcement learning agents by mimicking human-like task decomposition. As illustrated in Fig. 2, Dylan decomposes a complex goal into modular, logical subgoals (e.g., "collect the key" or "unlock the door") and assigns rewards through structured reasoning over these states. These rewards are semantically aligned with human intuition, enabling agents to learn more effectively in sparse-reward environments while maintaining high level of interpretability.

Beyond its role as a reward model, Dylan can also work as a planner that composes different behaviors by stitching together reusable logic options. This capability alleviates a key limitation of conventional RL agents: they are often overfit to a single task and fail to generalize. For instance, an agent trained to *navigate to a red door* may perform poorly when asked to *pick up a blue key and open a blue door*. Without modular reasoning and task composition, current RL systems must be retrained for every new task, incurring significant cost. Dylan's compositional structure supports generalization across related tasks and facilitates knowledge transfer through its symbolic task grounding. Moreover, Dylan's differentiable nature overcomes a common limitation of traditional symbolic planners, which are typically non-adaptive and prone to failure in environments requiring flexible search strategies. Overall, we make the following three major contributions:

**(i)** We propose Dylan, to the best of the authors' knowledge, the first differentiable symbolic planner.

**(ii)** Dylan can be integrated as a reward model into reinforcement learning frameworks, offering interpretable intermediate feedback that guides agents with fewer interactions while remaining aligned with human intent.

**(iii)** Beyond serving as a reward model, Dylan can also serve as a differentiable planner. Acting as a high-level policy in a hierarchical RL setting, it composes (logic) options in a modular and flexible way, allowing the agent to generate new behaviors without training.

## 2. Problem Setting and Related Work

Dylan builds upon several research areas, including first-order logic, differentiable reasoning, reinforcement learning and reward shaping. We start by introducing the problem setting: We define a **Symbolic Augmented Markov Decision Process (SA-MDP)** as a tuple:

$$\mathcal{M} = (\mathcal{S}, \mathcal{A}, \mathcal{R}, \rho_0, P, \mathcal{Z}, \mathcal{G}, \phi, \mathcal{C}),$$

where: $\mathcal{S}$ is the set of environment (low-level) states; $\mathcal{A}$ is the set of actions; $\mathcal{R} : \mathcal{S} \times \mathcal{A} \to \mathbb{R}$ is the low-level envi-

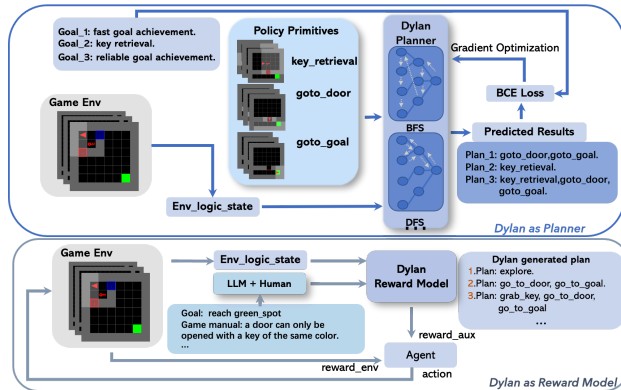

*Figure 2.* **Overview of Dylan as Planner (top) and Reward Model (bottom). As a differentiable symbolic planner**, Dylan is capable of adapting to tasks that require different search strategies and stitching together different (logic) options to generate novel behaviours. **As a reward model,** given the goal and game rules, Dylan first generates candidate plans to achieve the goal and then shapes rewards based on the generated plans, balancing among all candidate plans (as adaptive reward model) or selecting the best one to follow through (as static reward model).

ronment reward function; $\rho_0$ is the initial state distribution over $\mathcal{S}$; $P : \mathcal{S} \times \mathcal{A} \to \mathcal{P}(\mathcal{S})$ is the transition function; $\mathcal{Z}$ is the set of symbolic states, typically derived via an abstraction function $\phi : \mathcal{S} \to \mathcal{Z}$; $\mathcal{G} \subseteq \mathcal{Z}$ defines the symbolic goal conditions; $\mathcal{C}$ is the set of symbolic rules that encode domain knowledge and govern high-level planning or logic options $\mathcal{O}$. Given this SA-MDP setting, we consider the two core challenges: **Policy training acceleration:** leverage the symbolic structure (e.g., rules, goal conditions, abstractions) within the task to *speed up training the policies* through reward shaping. **Test time zero-shot generalization:** enable the agent to *compose new policies or behaviors at test time* by recombining previously learned logic options $\mathcal{O}$ or plans to solve novel tasks not encountered during training.

**First-order logic and Differentiable Forward-Chaining Reasoning.** We refer readers to App. I for a review of first-order logic fundamentals. Build upon this foundation, differentiable forward-chaining inference (Evans & Grefenstette, 2018; Shindo et al., 2021; Ye et al., 2022) enables logical entailment to be computed in a differentiable manner using tensor-based operations, which bridges symbolic reasoning with gradient based learning, allowing logic driven models to be trained end-to-end. Dylan builds upon these by introducing differentiable meta-level reasoning within the forward reasoning framework (Shindo et al., 2023), enabling symbolic planning in a fully differentiable architecture. Unlike classical symbolic planners (Fikes & Nilsson, 1971; 1993), Dylan learns to adaptively compose and select rules based on task demands, thus alleviating the non-adaptivity limitation of previous classical symbolic planners.

**Deep/ Neural-Symbolic Reinforcement Learning.** De-

spite recent advances in hierarchical RL (Nachum et al., 2018), meta-learning (Finn et al., 2017), and model-based RL (Kaiser et al., 2020), Deep RL algorithms still struggle with sample inefficiency, reward sparsity, and limited generalization. While neuro-symbolic methods like NLRL (Jiang & Luo, 2019), Galois (Cao et al., 2022), ESPL (Guo et al., 2023), and BlendRL (Shindo et al., 2025) integrate logic with RL, they primarily focus on synthesizing logical policies that map directly to low-level actions, whereas Dylan operates over logical options as a planner. Unlike model-based reinforcement learning (Moerland et al., 2023; Wu et al., 2022), the objective of Dylan is not to construct a world model via the logic planner. Instead, Dylan leverages prior knowledge encoded in the logic planner to shape the agent's behavior through informed reward signals, thus guiding and accelerating the learning process, rather than modeling environment dynamics. Recent works have shown trajectory stitching and generalization from demonstrations using diffusion models (e.g., DiffStitch (Li et al., 2024)) or learned representations (e.g., TMD (Myers et al., 2025), HQRL (Ke et al., 2025)). Dylan differs in how it achieves generalization. Rather than relying on trajectory embeddings, it leverages differentiable symbolic planning to modularly decompose and recompose tasks through logic options.

**Reward Shaping.** Reward shaping has long been studied as a means to improve sample efficiency and stability in RL, especially under sparse or delayed feedback. Classical potential-based methods (Ng et al., 1999) preserve policy invariance but rely on hand-crafted heuristics, limiting their expressiveness and interpretability. Extensions to multi-agent settings (Devlin et al., 2011) demonstrate utility but still lack compositionality. Reward Machines (Icarte et al., 2018; 2022) extend shaping with automata-based structures, offering more expressiveness but remaining static and hand-specified. Recent work explores unsupervised auxiliary tasks (Jaderberg et al., 2017) or preference-based learning (Christiano et al., 2017b) to ground rewards in richer signals. However, these approaches remain task-agnostic, sample-inefficient, or opaque. Likewise, intrinsic-motivation methods such as ICM (Pathak et al., 2017) and RND (Burda et al., 2019) focus on novelty or curiosity driven exploration rather than leveraging task structure. In contrast, Dylan introduces a differentiable symbolic planner as a reward model, enabling dynamic, structure-aware reward shaping. By providing modular and interpretable subgoal decomposition, Dylan aligns policy learning directly with task semantics.

## 3. Dylan: Guiding RL using Differentiable Symbolic Planning

A key feature of Dylan is its ability to incorporate human prior knowledge into (deep) RL in a structured and action-

able end-to-end fashion. To this end, Dylan uses (differentiable) symbolic logic to represent prior knowledge and guide the agent's through reward shaping. Let us illustrate this using a navigation task from the MiniGrid-DoorKey environment (Chevalier-Boisvert et al., 2023), where the agent must pick up a key, unlock a door, and reach the goal. By consulting the environment's manual, we extract high-level rules like: "a door can only be opened with a key of the same color" and "the agent can reach the goal only by passing through an open door." Instead of representing such rules as raw text or hardcoded logic, Dylan uses them as structured symbolic transitions: each step describes an action (e.g., go to the key) that transforms one state (e.g., no key) into another (e.g., has key), provided certain preconditions are met. These transitions are represented in a format inspired by STRIPS (Fikes & Nilsson, 1971), commonly used in classical planning. This symbolic abstraction allows Dylan to build a high-level plan that sequences primitive actions to achieve a given goal. To obtain task-specific rules as structured symbolic transitions, we prompt GPT-4o (OpenAI, 2024) to generate and transform candidate game rules, and then rely on human supervision to verify and refine them. We provide the detailed environment logic, game rules, the logic planner and the prompt format in App. B.

**Differentialize a symbolic planner.** By defining the initial and $t$-th step valuation of ground atoms as $\mathbf{v}^{(0)}$ and $\mathbf{v}^{(t)}$, we make the symbolic planner differentiable in three steps: **(Step 1)** We encode each planning rule $C_i \in \mathcal{C}$ as a tensor $\mathbf{I}_i \in \mathbb{N}^{G \times S \times L}$, where $S$ is the maximum number of possible substitutions for variables, $L$ is the maximum number of body atoms and $G$ is the number of grounded atoms. Specifically, the tensor $\mathbf{I}_i$ stores at position $[j, k, l]$ the index (0 to $G - 1$) of the grounded atom that serves as the $l$-th body atom when rule $C_i$ derives grounded head $j$ using substitution $k$. **(Step 2)** To be able to learn which rules are most relevant during forward reasoning, a weight matrix $\mathbf{W}$ consisting of $M$ learnable weight vectors, $[\mathbf{w}_1, \ldots, \mathbf{w}_M]$, is introduced. Each vector $\mathbf{w}_m \in \mathbb{R}^C$ contains raw weights for the $C$ planning rules. To convert these raw weights into normalized probabilities for soft rule selection, a *softmax* function is applied independently to each vector $\mathbf{w}_m$, yielding $\mathbf{w}_m^*$. **(Step 3)** At each step $t$, we compute the valuation of body atoms using the *gather* operation over the valuation vector $\mathbf{v}^{(t)}$, looping over the body atoms for each grounded rule. These valuations are combined using a soft logical AND (*gather* function) followed by a soft logical OR across substitutions: $b_{i,j,k}^{(t)} = \prod_{1 \leq l \leq L} \mathbf{gather}(\mathbf{v}^{(t)}, \mathbf{I}_i)[j, k, l]$, $c_{i,j}^{(t)} = softor^\gamma(b_{i,j,1}^{(t)}, \ldots, b_{i,j,S}^{(t)})$.

Here, $i$ indexes the rule, $j$ the grounded head atom, and $k$ the substitution applied to existentially quantified variables. The resulting body evaluations $c_{i,j}^{(t)}$ are weighted by their assigned rule weights $w_{m,i}^*$, and then aggregated across

rules and rule sets:$h_{j,m}^{(t)} = \sum_{1 \leq i \leq C} w_{m,i}^* \cdot c_{i,j}^{(t)}$, $r_j^{(t)} = softor^\gamma(h_{j,1}^{(t)}, \ldots, h_{j,M}^{(t)})$, $v_j^{(t+1)} = softor^\gamma(r_j^{(t)}, v_j^{(t)})$. We provide full details in App. C.

### 3.1. Dylan as Reward Model

**Dylan as static reward model**. With Dylan at hand, we now incorporate it into reinforcement learning and use it as a static reward model. The logical state is provided by the environment and used as an input to Dylan. Based on the state, goal and rules, Dylan generate a set of plans to reach the goal. Specifically, as a static reward model, Dylan selects the optimal *plan*, the one with the highest estimated probability of achieving the desired goal. Once the plan is determined, reward distribution follows the plan during decision-making.

Let the selected action sequence be denoted as $[a_1, a_2, \ldots, a_n]$, where each action $a_i$ represents a high-level action. Unlike the low-level actions commonly used in reinforcement learning, these high-level actions encode semantically meaningful behaviors, such as get_key. For each action, the planner also receives the corresponding expected state transition, represented as $\texttt{move}(a_i, s_{pre}^{(i)}, s_{post}^{(i)})$, where $s_{pre}^{(i)}$ and $s_{post}^{(i)}$ denote the pre- and post-action symbolic states, respectively as introduced previously. The **reward function** is designed to provide feedback only when the agent follows the planned action sequence in the correct order. Specifically, the agent must achieve each planned post-condition state $s_{post}^{(i)}$, corresponding to action $a_i$, before proceeding to the next action in the sequence. No reward is given if the agent deviates from the prescribed order or reaches states out of sequence. The auxilary reward function $r_{reasoner}(s, a, i)$ at i-th transition is defined as:

$$r_{reasoner}(s,a,i) = \begin{cases} \max\left(\lambda - \dfrac{n_{steps}}{N_{steps}}, 0\right), \text{if } s = s_{post}^{(i)}, \\ 0, \text{ otherwise.} \end{cases}$$

(1)

Here $\lambda > 0$, is a hyperparameter which denotes the reward. Dylan ensures the rewards are assigned **sequentially**, strictly following the planned order of actions $[a_1, a_2, \ldots, a_n]$. The agent must complete each transition $\texttt{move}(a_i, s_{pre}^{(i)}, s_{post}^{(i)})$ before progressing to the next step i+1. No rewards are granted if the agent skips steps, performs actions out of order, or transitions to incorrect states. Additionally, the reward is penalized by the number of steps taken num_steps relative to the total allowed steps total_steps. This incentivizes the agent to follow the planned sequence as efficiently as possible and discourages unnecessary actions. The shaped reward is defined as the sum of the environment reward and the reasoner reward:

$$r'(s,a,i) = r(s,a)_{env} + r_{reasoner}(s,a,i) \quad (2)$$

The objective becomes maximizing the expected cumulative discounted shaped reward:

$$J'(\pi) = \mathbb{E}_{\tau \sim \pi}\left[\sum_{t=0}^{\infty} \gamma^t \left(r(s,a)_{env} + r_{reasoner}(s,a,i)\right)\right]$$

(3)

By integrating the planner as a reward model, we provide structured and goal-aligned shaped rewards. This can lead to faster convergence and fewer training interactions, as shown empirically in the experimental section.

**Dylan as adaptive reward model.** We now introduce Dylan as an adaptive reward model. In contrast to static reward model, which only stick to the most promising plan, adaptive reward model leverages probabilities of all possible plans from Dylan. Specifically, building on the reward model described in Sec. 3.1, we introduce an additional dense reward $r_{adaptive}$ for each action. This reward encourages the agent to take steps that more effectively lead toward subgoals, thereby further accelerating learning process. The dense reward $r_{adaptive}$ is computed using the *log-sum-exp* (Cuturi & Blondel, 2017) over the probabilities of all candidate plans, ensuring numerical stability while leveraging the full distribution of plan probabilities. To obtain these probabilities, Dylan performs reasoning at each step of the agent's trajectory, producing updated likelihoods for each plan. The probability of each plan is obtained by $p_{targets} = \mathbf{v}^{(T)}\left[I_{\mathcal{G}}(\text{targets})\right]$, where $I_{\mathcal{G}}(x)$ returns the index of the target atom within the set $\mathcal{G}$. $\mathbf{v}^{(T)} = f_{infer}(\mathbf{v}^{(0)})$ denotes the valuation tensor obtained after $T$-step forward reasoning. $\mathbf{v}[i]$ represents the $i$-th entry of the valuation tensor. Details on how $\mathbf{v}^{(0)}$ is initialized are in App. J.

We aggregate the probabilities of all plans using the *log-sum-exp* (Cuturi & Blondel, 2017) operation for numerical stability, resulting in the dense reward:

$$r_{adaptive} = \log\left(\sum_{p_j \in \mathcal{P}_{a_t}^{(i)}} \exp\left(\log(p_j)\right)\right).$$

To ensure that the agent's learning is not dominated solely by the dense reward, we scale it by a factor $\omega$. As positive dense rewards could discourage the agent from finishing an episode by entering zero-reward absorbing states, we subtract a positive constant $\lambda$ ensuring that the dense auxiliary reward is always negative. The sparse rewards, discussed in Sec. 3.1 does not suffer from such survival bias, as it only given when the agent makes progress towards the goal. The resulting reward function, which integrates the adaptive reward with both the environment reward and the reasoner reward described in Sec. 3.1, is defined as:

$$r'(s,a,i) = \begin{cases} \omega\, r_{adaptive} - \lambda + r_{env} + r_{reasoner}(s,a,i), \\ \qquad\qquad\qquad \text{successful transition}, \\ \omega\, r_{adaptive} - \lambda + r_{env}, \quad \text{otherwise}. \end{cases}$$

(4)

Instead of sticking to a single fixed plan, our adaptive reward model takes all possible plans into consideration, guiding the agent to choose actions aligned with high-probability, goal-achieving strategies while discouraging stagnation or ineffective behavior.

### 3.2. Dylan as Differentiable Planner

Beyond its role as a reward model, Dylan can function as a standalone differentiable planner capable of sequencing primitive policies at execution time. By modularly "stitching" these policies together, it enables agents to complete novel tasks through zero-shot composition without additional training. Moreover, Dylan's differentiability alleviates the non adaptability of classical symbolic planners, allowing it to adapt its search strategy to meet the specific demands of a given task. To achieve this, Dylan employs a rule weight matrix $\mathbf{W} = [\mathbf{w}_1, \ldots, \mathbf{w}_M]$ to dynamically select planning rules. By applying a *softmax* function to each weight vector $\mathbf{w}_j \in \mathbf{W}$, we choose $M$ rules from a total of $C$ rules. The weight matrix $\mathbf{W}$ is initialized randomly and optimized via gradient descent, minimizing a Binary Cross-Entropy (BCE) loss between target (the desired final state) probability $p_{\text{target}}$ and the predicted probability $p_{\text{predicted}}$:

$$\underset{\mathbf{W}}{\texttt{minimize}} \quad \mathtt{L_{loss}} = \text{BCE}(p_{\text{target}}, p_{\text{predicted}}(\mathbf{W})). \quad (5)$$

Where the predicted probability $\mathbf{p}_{\text{predicted}} = \mathbf{v}^{(T)}[I_{\mathcal{G}}(\text{target})]$, $I_{\mathcal{G}}(x)$ returns the index of the target atom within the set $\mathcal{G}$. $\mathbf{v}^{(T)}$ denotes the valuation tensor obtained after $T$-step forward reasoning. $\mathbf{v}[i]$ represents the $i$-th entry of the valuation tensor. Due to its differentiability, Dylan can dynamically adapt its search strategies based on the tasks, which we demonstrate in Sec. 4.

> **Technical contributions summary:** Dylan's contribution lies in how differentiable planning is tightly integrated into the RL pipeline, both in terms of training signals and execution time compositionality. Concretely, Dylan: (1) provides structured, goal-aware reward shaping that guides the learning and exploration for Deep RL agents; and (2) enables differentiable symbolic option composition, allowing agents to solve multi-step tasks that are difficult for Deep RL agents under sparse feedback and long horizons.

## 4. Experimental Evaluation

In this section, we evaluate Dylan in two complementary roles: as a reward model and as a planner. **Dylan as a reward model.** We investigate whether incorporating "human" prior knowledge through differentiable symbolic planning improves reinforcement learning performance. Specifically, we ask: **RQ1:** Does Dylan enable agents to learn more

effectively with fewer environment interactions? **RQ2:** Is Dylan effective even when only partial symbolic knowledge is available? **RQ3:** Can adaptive reward shaping via Dylan further improve learning performance? **Dylan as a planner.** We assess Dylan's ability to generalize and adapt in planning-based settings. We ask: **RQ4:** Can Dylan adapt to tasks requiring different symbolic search strategies? **RQ5:** Can Dylan compose policy primitives to solve new tasks without retraining?

**Environment setup.** To evaluate and compare different reinforcement learning methods, we conduct a series of experiments within the MiniGrid (Chevalier-Boisvert et al., 2023), BabyAI (Chevalier-Boisvert et al., 2019) and (object-centric) Atari Learning Environment (Bellemare et al., 2013; Delfosse et al., 2024). MiniGrid and BabyAI offer a range of partially observable, grid-based tasks that are designed to test an agent's generalization and exploration capabilities. For consistency and reproducibility, we focus on DoorKey ($16 \times 16$, $12 \times 12$, $8 \times 8$), UnlockPickup (2 rooms), Key-InBox (9 rooms) and Freeway(Atari) environments. While the original DoorKey environment offers only a single viable solution path, we customized the DoorKey environments (as shown in Fig. 2) to support multiple solution paths, which provides two solutions for the agent to reach the goal position (indicated by the green marker). The agent can either directly go through an already opened blue door or alternatively acquire a red key, use it to unlock a red door, and navigate to the goal.Each environment presents different exploration challenges and sparse reward settings, making them ideal benchmarks for evaluating learning performance.

**Baseline methods.** We compare our approach against two sota neural baselines A2C (Mnih et al., 2016), PPO (Schulman et al., 2017), one hierarchical option framework baseline hDQN (Kulkarni et al., 2016), a logic baseline Galois (Cao et al., 2022) and a reward shaping baseline RND (Burda et al., 2019). Unless otherwise specified, hyperparameters are selected from the literature without fine-tuning (training hyperparameters in App. D and E).

**RQ1:** We evaluated Dylan's effectiveness as a static reward model for guiding RL agents during training. The primary goal of our experiments is to quantify how Dylan influences the convergence speed across different reinforcement learning algorithms. We conduct comparative evaluations using four benchmark environments: Unlock-Pickup (2 rooms), KeyInBox (9 rooms), customized DoorKey ($16 \times 16$ and $12 \times 12$) and Freeway (Atari). All environments (visualized in App. N ) emphasize the necessity for effective exploration due to their sparse reward setting.

Fig. 2 presents a detailed comparison of training performance across several reinforcement learning algorithms, specifically PPO and A2C, evaluated both with and without Dylan's auxiliary reward signals. Results are averaged over

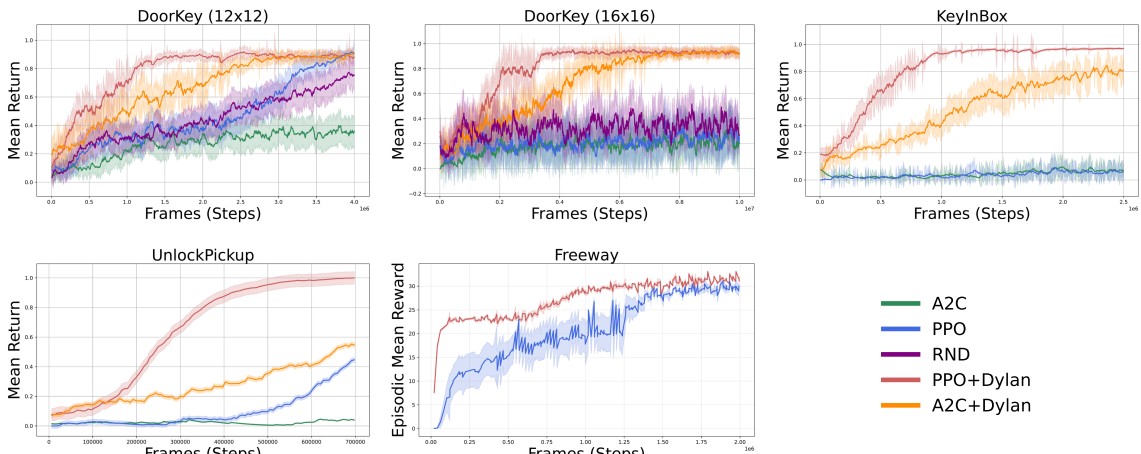

*Figure 3.* **As a static reward model, Dylan boosts RL agents' learning performance.** Comparisons of returns achieved by methods with and without Dylan in DoorKey($12 \times 12$, $16 \times 16$), KeyInBox (9 rooms), UnlockPickup (2 rooms) and Freeway(Atari) environments during training. The curves are averaged over ten runs (solid lines indicate mean values, shaded areas indicate standard error). All curves are smoothed using exponential moving average (EMA) for improved readability.

ten independent runs, with solid lines indicating mean performance and shaded regions denoting the standard error. For readability, all curves are smoothed using a time-based exponential moving average (EMA). Our empirical results clearly demonstrate Dylan's substantial positive impact on converge speed, particularly as the complexity of the environment increases. When assisted by Dylan all evaluated algorithms exhibit improved convergence rates in all environments. The improvement becomes notably greater in the challenging DoorKey ($16 \times 16$), KeyInBox (9 rooms), and Freeway (Atari). Remarkably, baseline agents such as PPO, which reliably converge in simpler settings such as DoorKey ($12 \times 12$), fail to converge in the challenging DoorKey ($16 \times 16$) environment without Dylan's guidance. In contrast, incorporating Dylan effectively resolves these exploration bottlenecks, significantly accelerating convergence. As a further illustration, we demonstrate that Dylan can provide guidance for continuous action spaces in MuJoCo MJX (Zakka et al., 2025) Quadruped navigation task (see App. M), however, we leave a detailed exploration of this domain for future work as it is outside the primary focus of this paper.

> **Takeaway 1:** As a static reward model, Dylan provides structured signals that boosts RL agents convergence speed. Functioning as a modular auxiliary reward provider, it delivers these gains without requiring modifications to the underlying RL algorithm.

**RQ2:** We further assess Dylan's ability to guide the agent's learning under imperfect situation (partial knowledge/ partial observability). The results in Tab. 2 show that Dy-

lan remains effective in accelerating policy learning even when operating with imperfect knowledge. In particular, Dylan_k, which lacks information about the key, struggles in the early stages but recovers rapidly after 300k steps, ultimately achieving performance comparable to the full-knowledge Dylan. Dylan_d, which omits door information, also demonstrates strong results, suggesting robustness to missing knowledge. Both Dylan variants consistently outperform the PPO baseline across most of the training trajectory, highlighting Dylan's ability to guide the agent effectively despite imperfect world knowledge. We provide additional experiments with missing rules in App. L.

> **Takeaway 2:** As static reward model, Dylan remains effective in accelerating Deep RL policy learning even when operating with imperfect knowledge.

**RQ3:** We further assessed Dylan's ability to guide the agent's learning by comparing two reward models (static and adaptive) alongside a baseline pure PPO approach. The static reward model delivers sparse rewards based on the agent's intermediate success, while the adaptive reward model provides denser and more informative rewards by dynamically evaluating the agent's progress. Fig. 4 illustrates the learning performance of different methods in the DoorKey ($8 \times 8$) environment, using both static and adaptive reward settings. Results are averaged over ten independent runs, with solid lines indicating mean returns and shaded regions denoting standard error. The results show that incorporating Dylan's auxiliary rewards consistently improves convergence speed compared to the pure PPO (Schulman et al., 2017) baseline. Notably, the adaptive

*Table 1.* **Dylan can generalize to unseen tasks through planning without retraining. Performance on multitasks (Success rate; the higher, the better).** We compare Dylan with the baseline method PPO (Schulman et al., 2017), A2C (Mnih et al., 2016), hDQN (Kulkarni et al., 2016), Galois (Cao et al., 2022), Goal-conditionned PPO (GC-PPO, following (Schaul et al., 2015; Hu et al., 2023)) and GPT-4o (Hurst et al., 2024) under perfect-options (and perfect-knowledge) assumptions (see Sec. 4, **RQ4** for details). Success rates are averaged over a hundred runs with their std. Best performing models are denoted using •. Models requires retraining for new goals are denoted using ◇.

| | Key Retrieval | Red Door Reaching | Goal Reaching | Safe Goal Reaching |
|---|---|---|---|---|
| A2C ◇ | $59.2_{\pm 12.5}$ | $50.2_{\pm 13.6}$ | $98.6_{\pm 1.6}$ | $41.2_{\pm 17}$ |
| PPO ◇ | $63.8_{\pm 12.2}$ | $53.2_{\pm 13.4}$ | $100_{\pm 0}$ • | $42_{\pm 17.2}$ |
| hDQN ◇ | $50_{\pm 0}$ | $35.8_{\pm 4.7}$ | $92.6_{\pm 3.8}$ | $46.8_{\pm 2.1}$ |
| Galois ◇ | $66.6_{\pm 4.9}$ | $68.4_{\pm 6.2}$ | $68.8_{\pm 8.5}$ | $27.4_{\pm 7.0}$ |
| GC-PPO ◇ | $99_{\pm 3.2}$ | $99_{\pm 3.2}$ | $99_{\pm 3.2}$ | $91_{\pm 8.7}$ |
| GPT-4o (with perfect assumption o) | $99_{\pm 3.2}$ | $97_{\pm 4.8}$ | $71_{\pm 12.9}$ | $99_{\pm 3.2}$ • |
| GPT-4o (with perfect assumption k+o) | $100_{\pm 0}$ • | $100_{\pm 0}$ • | $99_{\pm 3.2}$ | $98_{\pm 4.2}$ |
| Dylan (ours) | $100_{\pm 0}$ • | $100_{\pm 0}$ • | $100_{\pm 0}$ • | $98_{\pm 4.2}$ |

*Table 2.* **Dylan accelerates policy learning even with imperfect knowledge.** Performance on DoorKey ($8 \times 8$) with missing knowledge (return vs. training steps; higher is better). Dylan_k omits key visibility, Dylan_d omits door visibility. Despite incomplete knowledge about the world, Dylan with its variants outperform PPO. Results averaged over fifty runs with std.

| | 100k | 200k | 300k | 400k | 500k | 600k |
|---|---|---|---|---|---|---|
| PPO | $0.12_{\pm 0.24}$ | $0_{\pm 0}$ | $0.01_{\pm 0.03}$ | $0.19_{\pm 0.27}$ | $0.69_{\pm 0.18}$ | $0.85_{\pm 0.05}$ |
| Dylan | $0_{\pm 0}$ | $0.08_{\pm 0.18}$ | $0.65_{\pm 0.29}$ | $0.88_{\pm 0.04}$ | $0.88_{\pm 0.06}$ | $0.91_{\pm 0.04}$ |
| Dylan_k | $0_{\pm 0}$ | $0.07_{\pm 0.23}$ | $0.01_{\pm 0.03}$ | $0.55_{\pm 0.27}$ | $0.88_{\pm 0.06}$ | $0.86_{\pm 0.08}$ |
| Dylan_d | $0.05_{\pm 0.15}$ | $0.03_{\pm 0.06}$ | $0.3_{\pm 0.33}$ | $0.84_{\pm 0.07}$ | $0.87_{\pm 0.06}$ | $0.9_{\pm 0.02}$ |

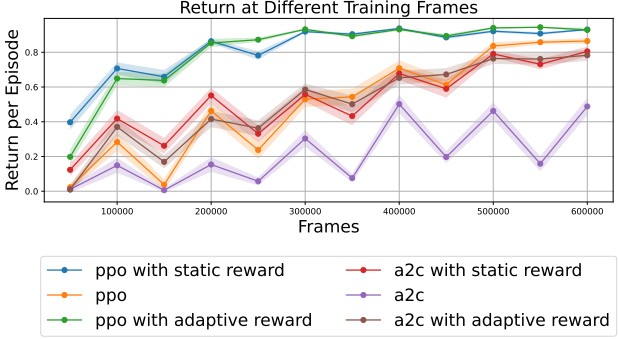

*Figure 4.* **Dylan further boosts RL agents' performances as an adaptive reward model**, shown by the returns over training on the DoorKey ($8 \times 8$) environment using PPO/A2C with static rewards, PPO/A2C with adaptive rewards, and pure PPO/A2C. All curves are averaged over ten runs; (solid lines indicate mean values, shaded areas indicate standard error).

reward model achieves slightly faster convergence than the static one, suggesting that accounting for multiple possible plans can further reduce training interactions. We provide the hyperparameters in App. K, full hyperparameter sweep results in App. P and wall-lock time comparison in App. Q.

> **Takeaway 3:** Dylan further enhances RL convergence speed as an adaptive reward model by accounting for the full distribution of possible plans.

We now demonstrate Dylan's planning capability to adapt its planning strategy to varying task structures and to com-

pose (logic) options in order to generate novel behaviors. Specifically, we evaluate two core capabilities: adaptive search strategy selection and compositional generalization.

**RQ4.** As a planner, Dylan's differentiability allows it to adapt its search strategy according to task demands. For example, in the scenario (task in App. H) Depth-First Search (DFS) may result in an infinite loop, where the agent aims to reach a state defined by `reach_goal` by combining two options: `go_through_red_door` and `go_through_blue_door`. However, when employing DFS, the recursive definitions of `get_through_door` may result in repeated exploration of the same paths. For instance, the agent may continuously attempt to execute the `go_through_red_door` or `go_through_blue_door` actions without progressing towards the final goal. This is particularly problematic when the agent encounters cyclic rules or tasks with high branching factors, where DFS tends to prioritize depth exploration over breadth, thereby getting stuck in loops or excessively deep branches. Under such conditions, the agent should autonomously recognize the inefficiency of DFS and accordingly shift to a more suitable method, such as Breadth-First Search (BFS). In this task (task provided in App. F), the planner is asked to solve each task within three reasoning steps. The first task is best addressed using DFS, while the second is more efficiently solved with BFS. To perform well across both scenarios, the agent must dynamically adjust its planning strategy to suit the structure of each task. Fig. 5 shows the training loss curves for Dylan on these two representative tasks (planner rules provided in App. G). All results are averaged over

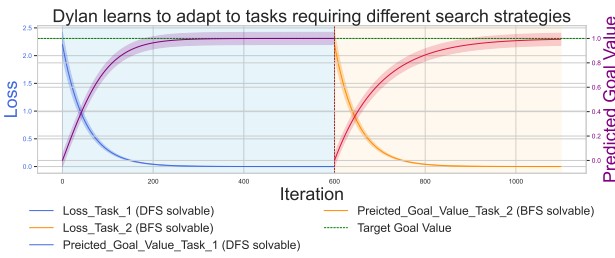

*Figure 5.* **As a differentiable symbolic planner, Dylan learns to adapt to tasks requiring different search strategies.** Loss curves and predicted goal value curve (target goal value 1.0) are averaged over five runs (solid lines= mean values, shaded areas = std).

five runs, with the solid line representing the mean and the shaded areas indicating the standard deviation.

> **Takeaway 4:** Dylan's differentiability allows it to adapt its search strategy to match the specific structural requirements of a task. Alleviating the non adaptability of classical symbolic planner.

**RQ5.** After demonstrating Dylan's differentiable adaptivity, we further evaluate its ability to compose options to produce novel behaviors, enabling the agent to solve previously unseen tasks without retraining. In this experiment, the planner is provided with a set of reusable logic options, such as `get_key` and `go_through_door`, and is asked to solve a range of goal-directed tasks using these building blocks. We designed four tasks in this experiment. In these tasks, the agent must retrieve a key, navigate to the red door, and reach the goal position while avoiding the blue door, which leads to a trap. Note that we use the customed DoorKey ($8 \times 8$) environment shown in Fig. 2, where the agent has multiple possible paths to reach the goal. As a result, possessing a key is not necessarily a prerequisite for opening a door.

For comparison, we include PPO, A2C, hDQN, GC-PPO, and two variants of GPT-4o as baselines. We implement GC-PPO by adapting PPO to a goal-conditioned variant following (Schaul et al., 2015; Hu et al., 2023). For GPT-4o (with po) and GPT-4o (with po+pk), the two settings differ in their assumptions. Perfect options (po) means 100% option success, which removes uncertainty introduced by the option layer. Perfect knowledge (pk) means access to the full knowledge (i.e., more information than Dylan received, we provide the prompt in App. O). Tab. 1 summarizes results across task variants. For each method, we evaluate 10 runs, each consisting of 10 episodes, and report the mean success rate with its standard deviation. Across these settings, Dylan successfully composes the correct sequence of logic options and generalizes to unseen tasks without training, whereas PPO, A2C, hDQN, and Galois (trained solely for goal navigation) fail on tasks requiring more complex compositional

behavior. While GC-PPO performs better than PPO, it requires retraining when the goal specification changes (e.g., when a new goal appears), whereas Dylan can reuse the same options and adapt whenever the new task is feasible. Although GPT-4o (with po) shows a slight improvement on the safe goal-reaching task, this gain stems from the perfect option assumption. By contrast, Dylan produces a 100% correct option sequence, and its remaining failures stem from option-layer uncertainty rather than planning errors. GPT-4o (with po+pk) achieves a success rate comparable to Dylan under the perfect assumption. Even under this favorable setting, GPT-4o does not reliably perform multi-step planning. Its outputs also require verification due to hallucinations (e.g., in a safe goal-reaching task, GPT-4o once suggested going through the trap (blue door)), and inference cost becomes prohibitive when frequent replanning is required. Note that all GPT-4o variants reported here operate without replanning. Dylan overcomes these limitations through a compact, differentiable symbolic planner that composes logic options on the fly to solve unseen, multi-step tasks.

> **Takeaway 5:** Dylan achieves zero-shot generalization by composing logic options to solve novel multi-step tasks without retraining. In contrast, RL baselines often require retraining for new tasks, while LLMs often fall short in logical reliability and replanning can be costly.

Overall, the experimental results provide affirmative answers to all five questions **Q1–Q5**.

## 5. Conclusions

In this paper, we introduced Dylan, a novel reward-shaping framework that leverages human prior knowledge to help reinforcement learning agents learn with less training interactions. We show, even with partial knowledge, Dylan significantly enhances RL agent's performance while reducing the required environment interactions. Beyond its role as a reward model, Dylan is, to the best of the authors' knowledge, the first differentiable symbolic planner that alleviates traditional symbolic planner's non-adaptable limitation. By dynamically composing logic options, Dylan enables the synthesis of novel, complex behaviors in a zero-shot fashion. Our empirical evaluations confirm Dylan's effectiveness in accelerating convergence and overcoming exploration bottlenecks, particularly as environment complexity increases. Ultimately, these results highlight Dylan's potential as a robust bridge between symbolic reasoning and deep reinforcement learning. Due to space limits, we put the limitations and future work in App. A.

## Impact Statement

This paper presents Dylan, an approach whose goal is to advance the field of Machine Learning. Specifically, the differentiable symbolic planner (Dylan) may improve the reliability of AI by bridging the gap between abstract reasoning and autonomous learning. By integrating "human-like" priors into reinforcement learning, Dylan replaces opaque "black-box" trial-and-error with a transparent, task-oriented logic that breaks complex problems into understandable steps. This structured approach may not only accelerates the development of more capable AI with fewer data interactions but also helps to ensure that autonomous systems are safer and more predictable. Ultimately, this work fosters human-centric AI that is easier to audit and align with societal expectations in sensitive areas such as automated decision-making. There are more potential societal consequences of our work, none which we feel must be specifically highlighted here.

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

## A. Limitations and future work

Although Dylan has demonstrated impressive results in enabling agents to learn from less environment interactions and generalizing to new tasks, it also has certain limitations. One limitation lies in its reliance on symbolic states provided directly by the environment. In future work, we aim to explore the use of vision foundation models to extract symbolic representations directly from raw game images. Another limitation involves the generation of game rules by large language models (LLMs). Currently, we prompt the GPT-4o to produce game rules and rely on human supervision to verify and correct them. A promising direction for future research is to incorporate an automated error-correction mechanism—such as leveraging multiple LLMs in a multi-round discussion framework—to improve the accuracy and reliability of the generated rules.

Promising avenues for future research include automating the acquisition of symbolic abstractions, for example, through predicate invention. Another interesting direction is exploring Dylan's broader application in inverse reinforcement learning (IRL), as it enables differentiable reasoning to infer underlying goals from expert demonstration within IRL setting. Additionally, investigating Dylan's scalability and broader applicability across both symbolic and subsymbolic domains remains an interesting research direction. We further want to apply Dylan to multimodal scenarios, where it could leverage multimodal inputs to more effectively guide agent learning across diverse and complex environments.

## B. Symbolic transformation for Dylan

Prompt: read the game manual and return the game rules in first-oder logic format such as:

$$\texttt{get\_through\_door:- initial, go\_through\_red\_door.}$$
$$\texttt{get\_through\_door:- get\_through\_door, go\_through\_blue\_door.}$$
$$\texttt{get\_through\_door:- get\_through\_door, go\_through\_red\_door.}$$
$$\texttt{reach\_goal :- get\_through\_door, go\_to\_goal.}$$

Dylan leverages human prior knowledge to guide reinforcement learning. As the first step, this prior knowledge is converted into a structured, logic-based format. For example, consider the agent navigation task illustrated in Fig. 2, drawn from MiniGrid Doorkey environment (Chevalier-Boisvert et al., 2023), After reviewing the environment's manual, we extract a rule stating that a locked door can only be opened with a key of the corresponding color, and that the agent can reach the goal by passing through an opened door. This rule can be expressed as the logic program:

$$\texttt{get\_red\_key:-init, go\_red\_key.} \quad \texttt{go\_through\_door:-init, go\_blue\_door.}$$
$$\texttt{go\_through\_door:-get\_red\_key, go\_open\_red\_door.}$$
$$\texttt{reach\_goal:-go\_through\_door, go\_to\_goal.}$$

We categorize the atoms in these clauses into two types: **state atoms** and **policy atoms**. Within each clause, the **head** and the **first body atom** are considered state atoms, while the **second body atom** corresponds to a policy atom. Following the STRIPS-style representation (Fikes & Nilsson, 1971; 1993), we transform each clause into a single atom using the predicate:

$$\texttt{move/3/[action, pre\_condition, post\_condition].}$$

For example, the clause: `get_blue_key :- initial, go_blue_key.` is rewritten as one atom: `move(go_blue_key, initial, get_blue_key)`. In this transformation, the **head** ( `get_blue_key`) becomes the **postcondition**, the **first body atom** ( `initial`) serves as the **precondition**, and the **second body atom** ( `go_blue_key`) represents the **action**.

With this transformation at hand, we now a symbolic planner:

$$\texttt{plan(Start, New, Goal, [Act, Old\_stack]):-}$$
$$\texttt{move(Act, Old, New), condition\_met(Old, Current),}$$
$$\texttt{change\_state(Current, New), plan(Start, Current, Goal, Old\_stack).}$$
$$\texttt{plan\_final(Start, Goal, Move\_stack):-}$$
$$\texttt{plan(Start, Current, Goal, Move\_stack), equal(Current, Goal).}$$

The first rule establishes the recursive process for generating a plan. It selects an appropriate policy that transitions the agent from an old state to a new state, verifies the necessary conditions, updates the current state accordingly, and recursively continues the planning process until the goal is reached. The second rule defines the termination condition, ensuring that the planning process concludes once the agent's current state matches the desired goal state.

## C. Differentialize Symbolic planner

We now describe each step in detail on how to differentialize the symbolic planner.

**(Step 1) Encoding Logic Programs as Tensors.** To enable differentiable forward reasoning, Each meta-rule is transformed into a tensor representation for differentiable forward reasoning. Each meta-rule $C_i \in \mathcal{C}$ is encoded as a tensor $\mathbf{I}_i \in \mathbb{N}^{G \times S \times L}$, where $S$ denotes the maximum number of substitutions for existentially quantified variables in the rule set , and $L$ is the maximum number of atoms in the body of any rule. For instance, $\mathbf{I}_i[j, k, l]$ stores the index of the $l$-th subgoal in the body of rule $C_i$ used to derive the $j$-th fact under the $k$-th substitution.

**(Step 2) Weighting and Selecting Meta-Rules.** We construct the reasoning function by assigning weights that determine how multiple meta-rules are combined. (i) We fix the size of the target meta-program to be $M$, meaning the final program will consist of $M$ meta-rules selected from a total of $C$ candidates in $\mathcal{C}$. (ii) To enable soft selection, we define a weight matrix $\mathbf{W} = [\mathbf{w}_1, \ldots, \mathbf{w}_M]$, where each $\mathbf{w}_i \in \mathbb{R}^C$ assigns a real-valued weight. (iii) We then apply a *softmax* to each $\mathbf{w}_i$ to obtain a probability distribution over the $C$ candidates, allowing the model to softly combine multiple meta-rules.

**(Step 3) Perform Differentiable Inference.** Starting from a single apply of the weighted meta-rules, we iteratively propagate inferred facts across $T$ reasoning steps.

We compute the valuation of body atoms for every grounded instance of a meta-rule $C_i \in \mathcal{C}$. This is achieved by first gathering the current truth values from the valuation vector $\mathbf{v}^{(t)}$ using an index tensor $\mathbf{I}i$, and then applying a multiplicative aggregation across subgoals:

$$b_{i,j,k}^{(t)} = \prod_{l=1}^{L} \mathbf{gather}(\mathbf{v}^{(t)}, \mathbf{I}i)[j, k, l], \tag{6}$$

where the $\mathbf{gather}$ operator maps valuation scores to indexed body atoms:

$$\mathbf{gather}(\mathbf{x}, \mathbf{Y})[j, k, l] = \mathbf{x}[\mathbf{Y}[j, k, l]]. \tag{7}$$

The resulting value $b_{i,j,k}^{(t)} \in [0, 1]$ reflects the conjunction of subgoal valuations under the $k$-th substitution of existential variables, used to derive the $j$-th candidate fact from the $i$-th meta-rule. Logical conjunction is implemented via element-wise product, modeling the "and" over the rule body.

To integrate the effects of multiple groundings of a meta-rule $C_i$, we apply a smooth approximation of logical *or* across all possible substitutions. Specifically, we compute the aggregated valuation $c_{i,j}^{(t)} \in [0, 1]$ as:

$$c_{i,j}^{(t)} = softor^{\gamma}(b_{i,j,1}^{(t)}, \ldots, b_{i,j,S}^{(t)}), \tag{8}$$

where $softor^{\gamma}$ denotes a differentiable relaxation of disjunction. This operator is defined as:

$$softor^{\gamma}(x_1, \ldots, x_n) = \gamma \log \sum_{i=1}^{n} \exp(x_i/\gamma), \tag{9}$$

with temperature parameter $\gamma > 0$ controlling the smoothness of the approximation. This formulation closely resembles a softmax over valuations and serves as a continuous surrogate for the logical *max*, following the log-sum-exp technique commonly used in differentiable reasoning (Cuturi & Blondel, 2017).

**(ii) Weighted Aggregation Across Meta-Rules.** We compute a weighted combination of meta rules using the learned soft selections:

$$h_{j,m}^{(t)} = \sum_{i=1}^{C} w^{m,i} \cdot c^{(t)}i,j, \tag{10}$$

where $h_{j,m}^{(t)} \in [0,1]$ represents the intermediate result for the $j$-th fact contributed by the $m$-th slot. Here, $w^{m,i}$ is the softmax-normalized score over the $i$-th meta-rule:

$$w_{m,i}^* = \frac{\exp(w_{m,i})}{\sum_{i'} \exp(w_{m,i'})}, \quad w_{m,i} = \mathbf{w}_m[i].$$

Finally, we consolidate the outputs of the $M$ softly selected rule components using a smooth disjunction:

$$r_j^{(t)} = softor^\gamma(h_{j,1}^{(t)}, \ldots, h_{j,M}^{(t)}), \tag{11}$$

which yields the $t$-step valuation for fact $j$. This mechanism allows the model to integrate $M$ soft rule compositions from a larger pool of $C$ candidates in a fully differentiable way.

**(iii) Iterative Forward Reasoning.** We iteratively apply the forward reasoning procedure for $T$ steps. At each step $t$, we update the valuation of each fact $j$ by softly merging its newly inferred value $r_j^{(t)}$ with its previous valuation:

$$v_j^{(t+1)} = softor^\gamma(r_j^{(t)}, v_j^{(t)}). \tag{12}$$

This recursive update mechanism approximates logical entailment in a differentiable form, enabling the model to perform $T$-step reasoning over the evolving fact valuations. The whole reasoning computation Eq. 6-12 can be implemented using efficient tensor operations.

## D. Hyperparameter

*Table 3.* Summary of Training Hyperparameters

| Parameter | Training Parameters |
|---|---|
| `--epochs` | 4 (PPO optimization epochs per update) |
| `--batch-size` | 256 (Batch size for PPO updates) |
| `--frames-per-proc` | 128 (Frames per process before update) |
| `--discount` | 0.99 (Discount factor $\gamma$) |
| `--lr` | 0.0001 (Learning rate) |
| `--gae-lambda` | 0.95 ($\lambda$ for GAE) |
| `--entropy-coef` | 0.01 (Entropy regularization coefficient) |
| `--value-loss-coef` | 0.5 (Value loss coefficient) |
| `--max-grad-norm` | 0.5 (Gradient clipping norm) |
| `--optim-eps` | $1 \times 10^{-8}$ (Optimizer epsilon) |
| `--optim-alpha` | 0.99 (RMSprop alpha) |
| `--clip-eps` | 0.2 (PPO clipping parameter $\epsilon$) |
| `--recurrence` | 1 (Recurrent steps, LSTM if $> 1$) |
| `--text` | `False` (Enable GRU for text input) |

## E. hDQN architechture

As the official implementation of hDQN is not publicly released, we provide our own PyTorch implementation following the methodology described in the original paper. Our architecture consists of two key neural network components: the **MetaController** and the **ControllerQNetwork**. These models form a hierarchical structure where the MetaController selects subgoals or abstract directives, and the ControllerQNetwork executes primitive actions conditioned on those directives.

### E.1. MetaController

The `MetaController` is a multi-layer feedforward network responsible for high-level decision making. It takes as input a feature vector representing the environment state and outputs a latent code or subgoal.

ARCHITECTURE

- Input: Feature vector of dimension `in_features`

- Linear layer: Linear(`in_features` $\rightarrow 512$)

- Layer Normalization: LayerNorm(512)

- LeakyReLU activation with slope 0.01

- Dropout: $p = 0.1$

- Linear layer: Linear($512 \rightarrow 256$)

- Layer Normalization: LayerNorm(256)

- LeakyReLU activation with slope 0.01

- Output Linear layer: Linear($256 \rightarrow$ `out_features`)

OUTPUT

A latent vector or high-level action representation of dimension `out_features`.

### E.2. ControllerQNetwork

The `ControllerQNetwork` is a value-based deep Q-network that operates at the lower level. It estimates Q-values for primitive actions given the current state and optionally the selected subgoal.

ARCHITECTURE

- Input: Feature vector of dimension `input_dim`

- Linear layer: Linear(`input_dim` $\rightarrow 512$)

- Layer Normalization: LayerNorm(512)

- LeakyReLU activation with slope 0.1

- Dropout: $p = 0.1$

- Linear layer: Linear($512 \rightarrow 256$)

- Layer Normalization: LayerNorm(256)

- LeakyReLU activation with slope 0.1

- Output Linear layer: Linear($256 \rightarrow$ `output_dim`)

OUTPUT

A vector of Q-values for `output_dim` discrete actions.

## F. Differentiable Planning Task

In this appendix, we present two pathological tasks that require a planner to adapt its search strategy—specifically between Depth-First Search (DFS) and Breadth-First Search (BFS)—for efficient goal finding within three steps. These examples demonstrate how a fixed strategy may underperform depending on the search structure and goal location.

**Task 1** favors DFS, as the goal `plan(a, h)` lies deep along a single branch:

```
edge(a, c). edge(a, b). edge(a, d).
edge(b, e). edge(c, f). edge(d, g).
edge(e, h).
```

**Task 2** favors BFS, as the goal `plan(a, e)` is close to the root but may be delayed by DFS exploring deeper branches first:

```
edge(a, c). edge(a, b).edge(b, d).
edge(c, e). edge(d, f).
```

## G. Differentiable Planning rules

We define the planning behavior of two search algorithms—**DFS** and **BFS**—using logical rules. These rules describe how each algorithm explores the search space and identifies successful plans.

```
dfs(B, F, G, r(F, D)):-edge(E, F), dfs(B, E, G, D).
plan(B, G):-dfs(B, F, G, H), equal(F, G).
bfs(k(B, D), S, E):-findall(A, F), append(C, F, k(B, D)), bfs(k(A, C), S, E).
plan(S, E):-bfs(k(A, C), S, E), equalbfs(A, C, E).
```

## H. Task that DFS can fail

```
get_through_door:- initial,go_through_red_door.
get_through_door:- get_through_door,go_through_blue_door.
get_through_door:- get_through_door,go_through_red_door.
reach_goal:- get_through_door,go_to_goal.
```

## I. First-Oder Logic

In first-order logic, a term can be a constant, a variable, or a function term constructed using a function symbol. We denote an $n$-ary predicate p as $p/(n, [\mathtt{dt_1}, \ldots, \mathtt{dt_n}])$, where $\mathtt{dt_i}$ represents the data type of the $i$-th argument. An atom is an expression of the form $p(\mathtt{t_1}, \ldots, \mathtt{t_n})$, where p is an $n$-ary predicate symbol and $\mathtt{t_1}, \ldots, \mathtt{t_n}$ are terms. If the atom contains no variables, it is referred to as a ground atom, or simply a fact.

A literal is either an atom or the negation of an atom. We refer to an atom as a positive literal, and its negation as a negative literal. A clause is defined as a finite disjunction ($\vee$) of literals. When a clause contains no variables, it is called a ground clause. A definite clause is a special case: a clause that contains exactly one positive literal. Formally, if $A, B_1, \ldots, B_n$ are atoms, then the expression $A \vee \neg B_1 \vee \ldots \vee \neg B_n$ constitutes a definite clause. We write definite clauses in the form of $A :\text{-} B_1, \ldots, B_n$. where $A$ is the *head* of the clause, and the set $\{B_1, \ldots, B_n\}$ is referred to as the body. For simplicity, we refer to definite clauses as clauses throughout this paper. The forward-chaining inference is a type of inference in first-order logic to compute logical entailment (Russell & Norvig, 2009).

## J. initial valuation obtained for adaptive reward model

To obtain the initial valuation $v_0$ for the adaptive reward model, we initialize both the environment's logic state and the high-level action atoms. Since the environment logic states are externally provided, the only variable component is the valuation of the high-level action atoms. We assign these valuations based on the agent's distance to each subtask, using the inverse of the distance (plus a small positive constant) to produce a meaningful and smoothly varying probability. This design ensures that as the distance changes, the corresponding probability in the plan is updated accordingly.

In our adaptive reward model experiments, we define the valuation of each high-level action atom as $0.5 + \frac{1}{\text{distance}+2}$. The plan probability is then computed based on the initial valuation of both the action atoms and the environment atoms.

*Figure 6.* **As a reward model, Dylan successfully guides the quadruped to the goal position.** In the quadruped navigation task, the robot must reach the goal (green dot) in Room B starting from the initial position (white dot) in Room A. After training, the pure PPO agent fails to reach the goal, whereas under Dylan's guidance, the quadruped successfully navigates to the target.

## K. hyperparameters for adaptive reward model

$\lambda = 0.01 \qquad \omega = 1/20$

## L. Dylan with missing rules

To assess DYLAN's robustness in our experiment, we conducted an ablation study in which 1/3 of the planning rules were randomly corrupted or removed (e.g., removing preconditions). We replaced each corrupted rule with a dummy logic option, a placeholder option that does not contribute to successful task execution, while leaving the remaining rules and options intact. The results, shown in Tab. 4, demonstrate that DYLAN maintains graceful degradation under substantial symbolic noise.

*Table 4.* Performance on multitasks setting with missing rules (success rate with std).

|  | Key Retrieval | Red Door Reaching | Goal Reaching |
|---|---|---|---|
| A2C | $59.2_{\pm 12.5}$ | $50.2_{\pm 13.6}$ | $98.6_{\pm 1.6}$ |
| PPO | $63.8_{\pm 12.2}$ | $53.2_{\pm 13.4}$ | $100_{\pm 0}$ |
| hDQN | $50_{\pm 0}$ | $35.8_{\pm 4.7}$ | $92.6_{\pm 3.8}$ |
| Galois | $66.6_{\pm 4.9}$ | $68.4_{\pm 6.2}$ | $68.8_{\pm 8.5}$ |
| Dylan (corrupted rules) | $93_{\pm 11}$ | $90_{\pm 12}$ | $80_{\pm 26}$ |
| Dylan (original) | $100_{\pm 0}$ | $100_{\pm 0}$ | $100_{\pm 0}$ |

## M. Quadruped navigation task (continuous action space)

In this task, the quadruped agent must navigate from Room A (white dot) to a goal in Room B (green dot). While standard PPO consistently fails to reach the goal, Dylan's guidance substantially improves the success rate, demonstrating its utility in continuous state-action spaces. We emphasize that these results are intended as a proof of concept; because our current task rewards are intentionally simple and not carefully engineered for this domain, Dylan enhances performance without yet guaranteeing a 100% success rate. This experiment primarily serves to illustrate that Dylan can operate effectively in high-dimensional continuous environments. Representative evaluation frames of a successful trajectory are shown in Fig. 6.

# N. Visulization of environments

Fig.7, Fig.8 and Fig.9 show the environments corresponding to the Unlock-Pickup, Key-in-Box and Freeway tasks, respectively.

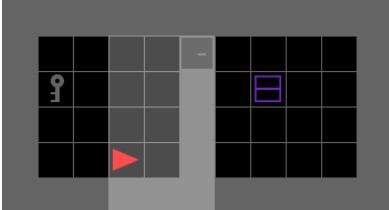

*Figure 7.* The UnlockPickup environment. The agent must first pick up the key to go through the door, then pick up the box. This game introducing a sequential dependency that increases the task's complexity.



*Figure 8.* The KeyInBox environment. The agent must first open the box to get the key, thenn using the key to go through the door. This game introducing a sequential dependency that increases the task's complexity.

# O. GPT-4o prompt

We provide the prompt we use for the GPT-4o expeirment, where *[image]* is an image and *[full symbolic state]* is the full symbolic state we provide to GPT-4o.

### O.1. Prompt for Perfect Knowledge and Perfect Option

Here is the image *[image]* and full symbolic state *[full symbolic state]*. The agent has access to four reusable logic options: `get_key`, `go_to_red_door`, `go_to_goal`, `go_to_blue_door`. Option preconditions and postconditions.`get_key`: precondition = *initial state*; postcondition = *key obtained*.`go_to_red_door`: precondition = *red key obtained*; postcondition = *go_through_the_door*. `go_to_blue_door`: precondition = *initial state*; postcondition = *go_through_the_door*. `go_to_goal`: precondition = *go_through_the_door*; postcondition = *goal reached*. We consider four tasks: 1. Key Retrieval: the goal is to obtain the key. 2. Red Door Reaching: the goal is to reach the red door. 3. Goal Reaching: the goal is to reach the goal position. 4. Safe Goal Reaching: the goal is to reach the goal position without crossing the blue door. generate now the option sequences for individual task.

### O.2. Prompt for Perfect Option assumption

Here is the image *[image]*. The agent has access to four reusable logic options: `get_key`, `go_to_red_door`, `go_to_goal`, `go_to_blue_door`. Your task is to generate a sequence of these four options listed, and finish the task: 1. Key Retrieval: the goal is to obtain the key. 2. Red Door Reaching: the goal is to reach the red door. 3. Goal Reaching: the goal is to reach the goal position. 4. Safe Goal Reaching: the goal is to reach the goal position without crossing the blue door. Now generate the option sequence for the four tasks for me.

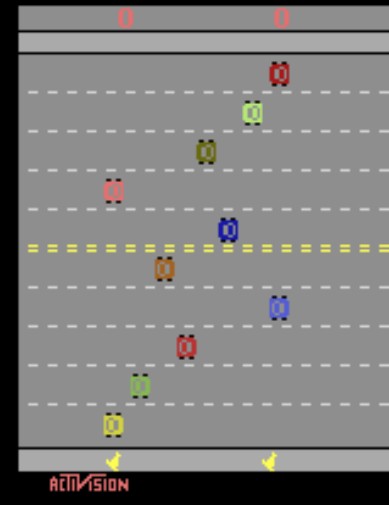

*Figure 9.* The Freeway environment. The agent need to cross the street to get rewards.

*Table 5.* DoorKey (12×12) training speed and wall-clock time comparison. Overall, adding Dylan or adaptive Dylan does not significantly increase the time cost relative to the corresponding baseline (A2C or PPO), while often reaching the success threshold in significantly fewer frames. In some settings (e.g., A2C), it can even enable convergence when the baseline fails to converge.

| Model | Frames | Time (s) | Converged | Converged at Frames | Converged at Time (s) |
|---|---|---|---|---|---|
| A2C | $1.9e7$ | 20407 | No | – | – |
| A2C+Dylan | $4e6$ | 11442 | Yes | $\approx 2.5e6$ | $\approx 7026$ |
| A2C+Dylan (adaptive) | $4e6$ | 16020 | Yes | $\approx 1.7e6$ | $\approx 6685$ |
| PPO | $4e6$ | 5418 | Yes | $\approx 4e6$ | $\approx 5418$ |
| PPO+Dylan | $4e6$ | 14186 | Yes | $\approx 2e6$ | $\approx 6695$ |
| PPO+Dylan (adaptive) | $4e6$ | 15818 | Yes | $\approx 1.5e6$ | $\approx 6014$ |

## P. Sweep hyperparameter

We provide the full hyperparameter sweeping results, including $\lambda, \omega$ in Fig. 10, and reasoning depth in Fig. 11.

## Q. Wall-clock Time comparison

Tab.5 summarizes the training time and convergence behavior on the DoorKey (12x12) task. For each method, we report the total number of environment frames, total training time, whether the run converged, and the convergence point, indicating frame time when applicable. All the experiments are done using Apple M3 chips with 16GB RAM.

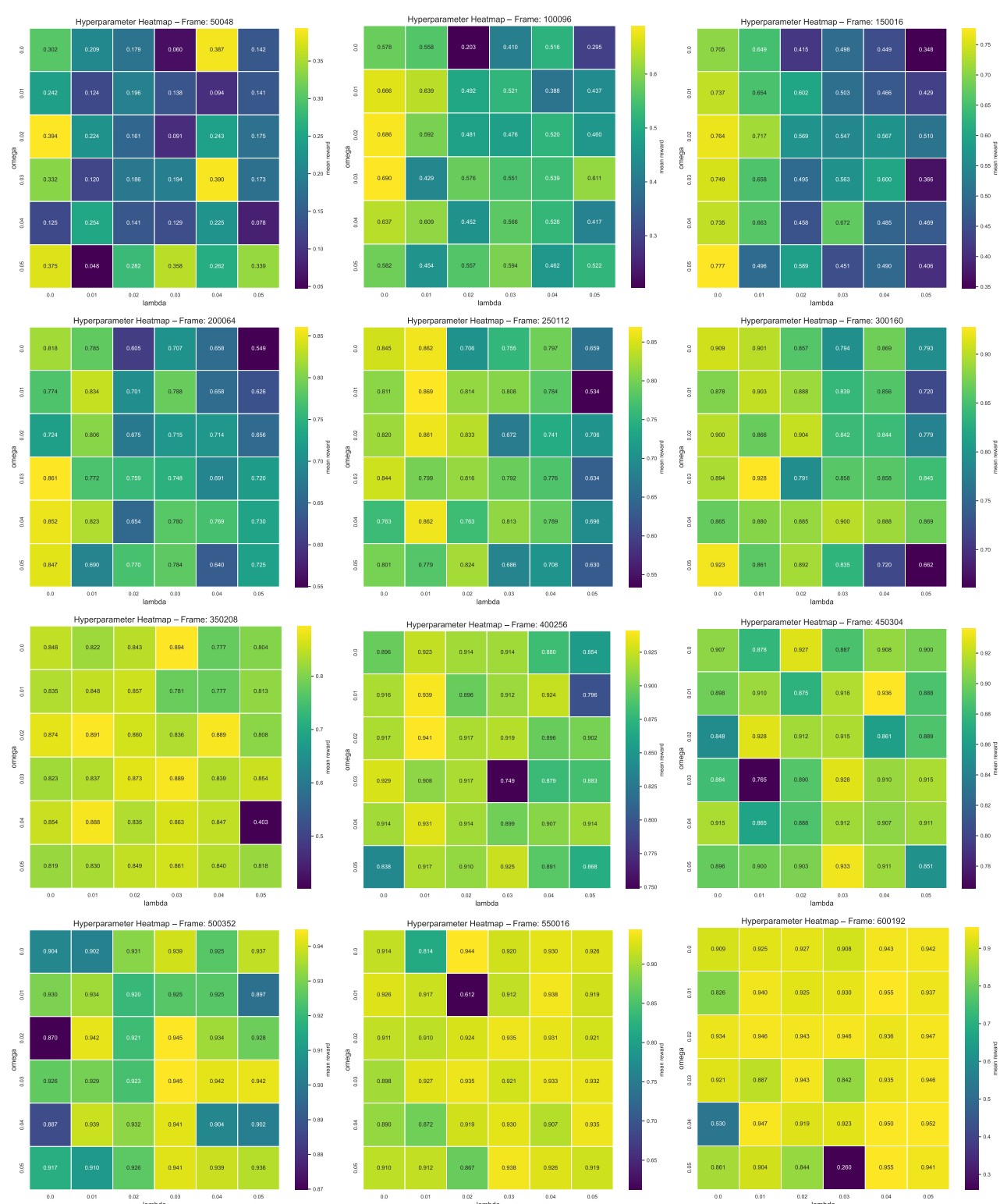

*Figure 10.* We sweep hyperparameters $\lambda, \omega$, and evaluate the episodic reward throughout training. Brighter regions indicate higher rewards.

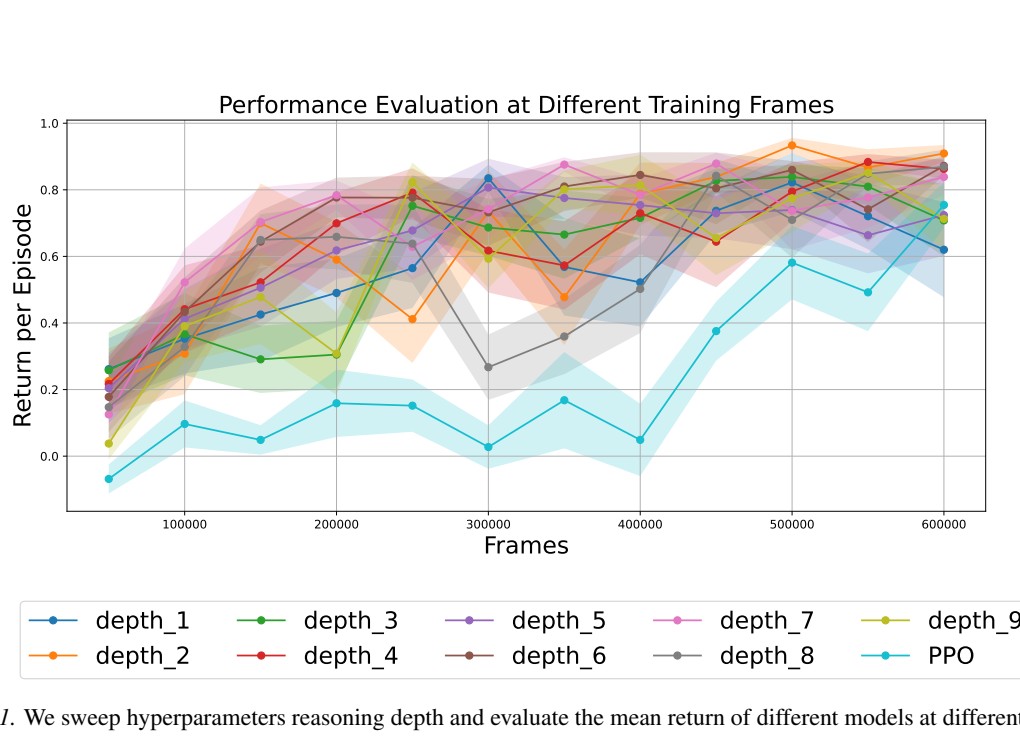

*Figure 11.* We sweep hyperparameters reasoning depth and evaluate the mean return of different models at different time steps.

