# OpenReview forum: "Learning from Less: Guiding Deep Reinforcement Learning with Differentiable Symbolic Planning"
_ICML.cc/2026/Conference — Submitted to ICML 2026_

### Official Review · Reviewer_NhCi · 2026-03-10

**Soundness:** 2
**Presentation:** 3
**Significance:** 3
**Originality:** 3
**Overall Recommendation:** 4
**Confidence:** 3

**Summary:**

This paper presents Dylan, a differentiable symbolic planner for sparse-reward reinforcement learning (RL) that leverages human priors as first-order logic to enable sample-efficient learning. Dylan can provide either static or dynamic reward signals to an RL agent based on the logical state, goal, and rules. It gives the agent a static reward if the current state matches the post-condition state of the best high-level plan, or an adaptive reward computed as the smooth maximum over generated plans. Furthermore, Dylan can serve as a high-level differentiable symbolic planner given a set of primitive policies, enabling test-time execution without further training. The authors evaluate Dylan across several domains that present challenges with sparse rewards: MiniGrid, BabyAI, Atari, and MuJoCo.

**Compliance With Llm Reviewing Policy:**

Affirmed.

**Final Justification:**

I will change the final recommendation score from "weak reject" to "weak accept" based on the belief that the authors will reflect the change in the rebuttal. The main reason for the increase in score is that the authors adequately addressed the raised concerns in their rebuttal. Although this paper constrains the experimental scope to a narrow toy environment (i.e., BabyAI), and I partially agree with the other reviewer's opinion, this paper could offer insight into the possibility of integrating a differentiable, human-readable planning procedure into the decision-making process for deep RL practitioners.

**Key Questions For Authors:**

- How do you define the similarity of tasks when performing the zero-shot generalization tasks?
- How do you compute `num_steps` and `total_steps` for the static reward model?
- What are the primitive policies that Dylan uses for selecting an action? Are they pre-trained vanilla RL methods (e.g., PPO) with a sub-task (or sub-goal) or human-engineered policies (e.g., select an action based on pre-defined rules)?
- According to Appendix K and P, I conjecture that the authors have chosen the same Dylan's hyperparameter across experiments based on the sweeping experiment. Since the reward signals from Dylan's reward models are scaled and shifted with the hyperparameters, how robust are Dylan's reward models to the choice of hyperparameters across different environments where the environmental reward is bounded? For instance, the DoorKey task in MiniGrid computes the final reward based on the formula—r_env = 1 - 0.9 * (`step_count` / `max_steps`). Since the maximum value of `step_count` is bounded by an optimal path depending on the size of the grid, the value of `r_env` is inherently bounded by the size of the grid. To prevent the dominant or minor effect of `r_env`, choosing such hyperparameters could affect the performance of Dylan critically.

**Limitations:**

Yes

**Strengths And Weaknesses:**

## Strengths
- The authors present a novel framework that incorporates a high-level differentiable symbolic planning with gradient-based policy learning in RL, which can easily integrate existing deep RL methods with symbolic, human-like reasoning capability.
- The authors rigorously explain the connectivity between prior works and Dylan, supporting the originality and novelty of the proposed method.
- The experimental setup is motivated by research questions, sequentially well-structured to help the readers understand Dylan's strengths.

## Weaknesses
- The authors claim that Dylan can assign static or adaptive rewards based on logical sub-goals (e.g., *collect the key"), which semantically aligns with the human intent (L64 ~ L65). To further verify whether Dylan's reward model indeed generates reward signals that align with the given logical states, the authors should perform an additional experiment comparing the predicted rewards from Dylan when two agents behave differently: one agent generates human-intent-aligned actions while the other one doesn't.
- The experiments are incompletely designed to fully validate the strengths of Dylan. I address my concerns in the following:
- - According to the authors, research question 1 (RQ1) aims to answer whether Dylan improves the convergence speed of underlying deep RL methods in a sparse-reward setting. Since Dylan relies on a backbone deep RL method to obtain low-level actions, we can compare the sample efficiency of several exploration strategies [1-4] in the sparse-reward environment, including Dylan's reward model. Considering the authors compare Dylan with its vanilla variants without exploration strategies (i.e., A2C and PPC) and RND, which is far outdated, it seems hard to clarify the clear benefits of Dylan in RQ1. Last but not least, inconsistent baselines in Figure 3 (e.g., No RND in KeyInBox and UnlockPickup; No A2C, A2C+Dylan, and RND in Freeway) exacerbate the reliability of experimental results.
- - In RQ2, inadequate descriptions of manipulating the missing knowledge for Dylan imply a question on how Dylan is sensitive to the portion of imperfect knowledge.
- - In RQ4, it is unclear how Dylan modifies (or switches) its search strategy automatically. Figure 5 may indicate the switching point depicted by a vertical red line in the middle, while the legends are not aligned with the plots. In addition, the title is included in the plot, which is not compatible with the template provided by ICML.
- - In RQ5, many details about the experimental setup are omitted.
- - - The extent to which Dylan is trained and reuses the trained behaviors is obscure to solve the task described in RQ5.
- - - The criteria that determine whether a specific task is seen or unseen are omitted, which clarifies the generalization toward the capability of unseen tasks.
- - - According to the manuscript, other baselines are either separately trained with goal-reaching tasks and possibly options or adopted with the same primitive policies and different high-level planners (i.e., GPT-4o). As the current manuscript lacks significant details in the training scheme and task decomposition (i.e., the way a complex, long-horizon task is divided into subtasks), it seems hard to verify the authors' claim that Dylan demonstrates strong zero-shot generalization without retraining.

## Minor problems
- Typos:
- - Build -> Building (L94)
- - ~ the goal.Each ~ -> ~ the goal. Each ~ (L248)
- - Fig. 2 -> Fig. 3 (L270)

### Reference
[1] Liu, Hao, and Pieter Abbeel. "Aps: Active pretraining with successor features." ICML 2021

[2] Yarats, Denis, et al. "Reinforcement learning with prototypical representations." ICML 2021

[3] Andres, Alain, Esther Villar-Rodriguez, and Javier Del Ser. "Towards improving exploration in self-imitation learning using intrinsic motivation." SSCI 2022

[4] Wan, Shanchuan, et al. "Deir: efficient and robust exploration through discriminative-model-based episodic intrinsic rewards." IJCAI 2023

---

> ### Author Rebuttal · Authors · 2026-03-30
>
> We thank the reviewer for the positive comments on novelty and appreciate the detailed suggestions for clarifying the setup. Here are the detailed responses.
>
> **Weakness:** :
> - Thank you for your suggestion. We additionally evaluate three agents: human-random, human-optimal, and the Dylan agent on an 8×8 maze with two possible paths and two doors, each of which may be locked or unlocked. When both doors are locked, the optimal strategy is to first obtain the key and then open the matching door. Otherwise, the optimal behavior is to go directly through the unlocked door without collecting the key. The random policy does not necessarily follow this optimal path but still finishes within a 30-step limit. Rewards are produced by Dylan (adaptive) and accumulated step by step. The results are averaged over three episodes, showing that Dylan agent aligns closely with the human-optimal policy.
>
> | Policy | accumulated reward with std |
> |---|---|
> | human random | 3.72 +/- 0.14 |
> | human optimal | 4.83 +/- 0.09 |
> | Dylan | 4.82 +/- 0.12 |
>
> - RQ1 is not intended to establish a new SOTA RL algorithm. Rather, its purpose is to demonstrate the effectiveness of logic-based guidance in improving the underlying RL agent. We therefore included baselines that are sufficient to support the corresponding claim. In the revision, we can add these additional methods to all environments.
>
> - RQ2 In the main text, partial observability is present in the original MDP, but not in the domain knowledge. We show imperfect domain knowledge in App L. The results show that Dylan remains effective even when the domain knowledge is incomplete: when one-third of the knowledge is missing, it still improves performance. However, as the amount of missing knowledge increases further, the framework eventually breaks down. Handling imperfect domain knowledge is not the focus of the current work; rather, we view it as an orthogonal and important direction for future investigation, and we will clarify this point in the revision.
>
> - RQ4: We apologize for the confusion. We have removed the title. The legend colors are  Predicted_Goal_Value_Task_1  in purple and Predicted_Goal_Value_Task_2  in red. The automatic adaptation is realized through gradient-based learning of the planning-rule weights. We provide the planner with the goal state and DFS- and BFS-style planning rules (as in App.G), with random initial weights. At step 600, the task changes from being DFS- to BFS-solvable, requiring the planner to adapt its strategy accordingly. The planner then updates the rule weights via gradient descent and learns which strategy is more appropriate for the task. We will improve our clarity in our revision.
>
> - RQ5: In this task, Dylan is not trained, rather, it is provided with domain knowledge and operates as a planner that generates a sequence of actions  to reach the target goal. The task decomposition performed by Dylan is encoded in the domain knowledge, such as the logic programs shown in App B. We will add a new appendix in the revision that explicitly presents the domain knowledge used by Dylan. We thank the reviewer for pointing out this need for clarification. For GPT-4o, the corresponding prompt details are provided in App O. Finally, we use seen to denote tasks whose policies are trained with the corresponding reward, and unseen to denote all other tasks
>
> **Q1:** We don't quantify task similarity. Instead, we evaluate the compared methods on multiple tasks that differ in their required subgoal composition and planning structure. Our goal is therefore to test whether the approach can generalize to new task compositions rather than to measure generalization as a function of a predefined task-similarity metric. We will clarify this in the revision. We notice that the term zero-shot may be too strong in our current presentation. While Dylan does not require retraining on the unseen tasks, it is still provided with domain knowledge for planning. For this reason, the more accurate description is test-time composition without policy retraining. We apologize for this imprecise wording and have revised it accordingly.
>
> **Q2:** *num_steps* is the step that the agent has taken. *total_steps* is the maximum number of steps the agent can take in one episode.
>
> **Q3:** They are PPO policies.
>
> **Q4:** We agree that the optimal hyperparameters of the reward function can depend on the specific environment. However, in our intended use case, the system would be deployed and optimized for a given environment, but should generalize over different tasks to enable autonomous, planning-guided exploration.

---

> > ### Author Rebuttal · Reviewer_NhCi · 2026-04-02
> >
> > Thank you for conducting an additional experiment during the short rebuttal period and for your detailed explanations. Regarding RQ2-5, the authors' descriptions and future revision plans successfully resolve my concerns. However, it is hard to be convinced by RQ1, in which the authors claim that the current experimental setup is sufficient to verify that Dylan enables sample-efficient RL via logic-based guidance. I agree that this paper does not intend to suggest a new SOTA RL method. That said, this paper should consider the same backbone RL method for policy learning (e.g., PPO) with different guidance methods, separating the guidance (i.e., exploration) from the policy learning (i.e., downstream RL) as much as possible. In my view, only two variants (i.e., A2C + Dylan and PPO + Dylan) are effective to support the claim in RQ1, while other guidance strategies for comparison are still missing. Furthermore, based on the authors' rebuttal, it seems the takeaways highlighted by purple boxes may overclaim Dylan's capability and over-generalize beyond the extent this paper can prove. If the authors can provide further results by considering other baselines for RQ1, I will raise my score. Otherwise, I will maintain my score.

---

> > > ### Author Response · Authors · 2026-04-05
> > >
> > > We thank the reviewer for engaging with the rebuttal, and we are pleased that we have addressed the concerns raised. Regarding RQ1, we ran 3 additional exploration baselines in the DoorKey 12x12 environments, including ICM[1], NGU[2], and NovelD[3], all using the same PPO as the underlying policy as PPO+Dylan. The detailed results are provided below. All experiments are averaged over three seeds and reported using the original environment reward; the best-performing models are bolded. We are currently running the additionally added baselines in other environments and will update our manuscript accordingly. The results show that Dylan offers clear benefits in terms of convergence speed compared with these exploration-based baselines. Moreover, Dylan provides a unique additional advantage: interpretable intermediate guidance. We will correct the imprecise wording in the takeaway box 5 and carefully review for further cases of overclaiming.
> > >
> > > ---------
> > > Update: We now include the fourth additional baseline, DEIR [4], in the table
> > >
> > > |Method\Steps |6400|326400|646400|966400|1286400|1606400|1926400|2246400|2566400|2886400|3206400|3526400|3846400|
> > > |-|-|-|-|-|-|-|-|-|-|-|-|-|-|
> > > |**A2C**|0.07±0.04|0.12±0.06|0.12±0.08|0.26±0.12|0.30±0.08|0.38±0.10|0.30±0.10|0.32±0.09|0.34±0.13|0.32±0.08|0.35±0.11|0.33±0.09|0.32±0.09|
> > > |**PPO**|0.09±0.06|0.14±0.04|0.28±0.08|0.31±0.09|0.40±0.11|0.38±0.11|0.39±0.11|0.33±0.11|0.52±0.09|0.59±0.03|0.72±0.08|0.79±0.09|0.86±0.01|
> > > |**PPO+Dylan**|0.19±0.03|**0.64±0.06**|**0.68±0.07**|**0.75±0.03**|**0.81±0.02**|**0.81±0.11**|**0.84±0.03**|**0.84±0.01**|**0.90±0.05**|0.86±0.02|**0.89±0.05**|**0.89±0.01**|**0.89±0.02**|
> > > |**A2C+Dylan**|**0.33±0.09**|0.45±0.08|0.56±0.06|0.64±0.06|0.67±0.08|0.70±0.07|0.73±0.09|0.76±0.07|0.84±0.02|**0.87±0.03**|0.87±0.04|0.85±0.01|0.86±0.02|
> > > |**RND**|0.03±0.03|0.18±0.08|0.29±0.11|0.31±0.09|0.32±0.08|0.34±0.11|0.39±0.12|0.43±0.12|0.51±0.13|0.59±0.11|0.62±0.10|0.64±0.13|0.71±0.11|
> > > |**ICM**|0.00±0.00|0.19±0.15|0.34±0.03|0.27±0.07|0.34±0.20|0.35±0.20|0.47±0.09|0.43±0.12|0.55±0.19|0.68±0.25|0.76±0.02|0.82±0.25|0.89±0.08|
> > > |**NGU**|0.21±0.17|0.26±0.21|0.28±0.09|0.25±0.23|0.41±0.21|0.31±0.13|0.43±0.19|0.36±0.08|0.40±0.23|0.55±0.07|0.58±0.05|0.76±0.04|0.87±0.03|
> > > |**NovelD**|0.00±0.00|0.24±0.20|0.24±0.24|0.41±0.15|0.30±0.06|0.34±0.13|0.39±0.10|0.46±0.03|0.47±0.22|0.62±0.18|0.71±0.01|0.81±0.06|0.88±0.01|
> > > |**DEIR**|0.05±0.04|0.21±0.02|0.27±0.03|0.27±0.25|0.35±0.14|0.34±0.07|0.35±0.06|0.46±0.22|0.67±0.10|0.76±0.02|0.77±0.05|0.83±0.04|0.84±0.26|
> > >
> > > [1] Pathak D, Agrawal P, Efros A A, et al. Curiosity-driven exploration by self-supervised prediction[C]//International conference on machine learning. PMLR, 2017: 2778-2787.
> > >
> > > [2] Badia A P, Sprechmann P, Vitvitskyi A, et al. Never Give Up: Learning Directed Exploration Strategies[C]//International Conference on Learning Representations, 2020.
> > >
> > > [3] Zhang T, Xu H, Wang X, et al. Noveld: A simple yet effective exploration criterion[J]. Advances in Neural Information Processing Systems, 2021, 34: 25217-25230.
> > >
> > > [4] Wan S, Tang Y, Tian Y, et al. DEIR: efficient and robust exploration through discriminative-model-based episodic intrinsic rewards[C]//Proceedings of the Thirty-Second International Joint Conference on Artificial Intelligence. 2023: 4289-4298.

---

### Official Review · Reviewer_8u2F · 2026-03-11

**Soundness:** 2
**Presentation:** 1
**Significance:** 2
**Originality:** 2
**Overall Recommendation:** 3
**Confidence:** 4

**Summary:**

The paper is about the use of symbolic, high-level planning knowledge for guide a RL agent that deals with low-level state representations. The symbolic knowledge is suppose to capture "human priors", allowing the agent to leran with fewer training interactions. Common benchmarks are used to evaluate the concrete proposal like BabyAI, Minigrid, and ALE. The symbolic knowledge takes the form of rules o STRIPS actions are they obtained from LLMS and with suitable manual intervention. The symbolic knowledge ends up playing several possible roles that are analyzed; as reward model for symbolic intermediate state achieved, as an "adaptive reward model" for rewards actions in the possible plans (if I understand correctly), and of course, as a high-level planner.

**Compliance With Llm Reviewing Policy:**

Affirmed.

**Final Justification:**

Interesting ideas on combining deep RL with symbolic knowledge that involves learnable parameters; but the idea of "symbolic differential planning" in the paper does not come out clear enough.

**Key Questions For Authors:**

Can you explain the notation and the formula that defines $r_{adaptive}$? What is actually ${\cal P}_{a_t}^i$? Where is this formula coming from?

Can you rephrase what is the differentiable symbolic planner and how it is used (parameters included) to generate the high-level plans and to deserve that name?

**Limitations:**

Yes

**Strengths And Weaknesses:**

A strength of the paper is that the experimental results show the difference that the extra symbolic knowledge can make in relation to plain RL settings. The weaknesses have to do with the lack of comparison with other approaches that use background, symbolic knowledge in a RL, and the convoluted description of the proposed approach. In particular, the "differentiable symbolic planner" is mentioned in the title, abstract, and as the first contribution, but it is described in one paragraph in page number 3, in a way that's cryptic and low level. Why call this a differentiable symbolic planner? In a sense, they seem to be parameterized rules, associated with STRIPS actions, but it's not clear from these description how the resulting weights are used in the planner that generates the high-level sequence of states. These are important parts of the paper, but their description is poor and not sufficiently precise, and doesn't make justice to the claim of a "different symbolic planner".

---

> ### Author Rebuttal · Authors · 2026-03-30
>
> We thank the reviewer for emphasizing the empirical value of using symbolic knowledge and for the suggestions for a clearer description. Here are our detailed replies to the weakness and questions.
>
> **Q1:** Thank you for bringing this up. We will revise the manuscript as follows: $r_{adaptive}$​ denotes the adaptive reward, which takes into account the probabilities of all candidate plans associated with the i-th transition action $a_t​$ using log sum exponential trick. Here, $P_{a_t}^i$​​ denotes the set of probabilities of the individual plans associated with the i-th transition with action $a_t$.  $p_j$ denotes the probability of a single plan. These probabilities are obtained from the differentiable planning process, in which each candidate plan is inferred with a probability at every step.
>
> **Q2:** The initial valuation v(0) of Dylan is a vector in the range [0,1] that contains the initial belief for each grounded atom, for example, [ move(get_key,initial,key_obtained), plan(initial,initial,goal,[ ]), … ]. Shindo et al. propose a differentiable mechanism for computing the valuation vector at the next reasoning step, e.g., v(1), which represents the valuations that can be derived from a given set of clauses. However, reasoning alone does not provide planning capability. One of our contributions is to extend this formulation to the planning setting. Specifically, we introduce meta-predicates such as *move/3* and *plan/4* (as illustrated in App B), together with two meta-clauses (*plan* and *plan_final*, also in App B), which enable us to derive a symbolic plan in a differentiable manner by recursively deriving the goal predicate. For example:
>
> 1.0::plan(Start, New, Goal, [Act, OldStack]) :-move(Act, Old, New), condition_met(Old, Current), change_state(Current, New), plan(Start, Current, Goal, OldStack).
>
> Starting from v(0), we iteratively apply the meta-clause *plan* until we obtain an action stack that connects the start state to the goal state. Once the goal is reached, the *plan_final* meta-clause is applied to verify that a valid solution has been found. More details are provided in App B and C. In App G, we also show how different meta rules can be used to realize DFS-style planning, BFS-style planning, or a combination of both. We will revise the paper to make this process clearer.
>
> To illustrate the value of differentiable planning, consider the case where the agent observes that a door is already open, so obtaining the corresponding key is no longer necessary. In that situation, the plan that directly passes through the door will receive a higher probability than the plan that first retrieves the key. As a result, going through the door becomes the preferred action. Because the entire computation is end-to-end differentiable, these plan probabilities can be used to guide the agent, as demonstrated by our adaptive reward: actions that increase the likelihood of task completion can be encouraged, while actions that reduce it can be discouraged.
>
> Moreover, because the entire pipeline is differentiable, we can learn the weights of the planning rules. As shown in RQ4, we provide the planner with the goal state together with DFS- and BFS-style planning rules whose weights are randomly initialized. The planner then adapts these rule weights through gradient-based learning to determine which strategy, BFS or DFS, is better suited for solving a given task.
>
> **Weakness:** These comparisons were chosen to cover both end-to-end RL methods and hybrid symbolic-RL frameworks. As we introduced in paper line 114-116, while we are aware of recent neuro-symbolic RL methods such as ESPL [1], NLRL [2] and BlendRL [3]. These works differ in focus, they focus on learning symbolic or logical policies that map directly to raw actions. In contrast, Dylan operates at the level of symbolic options. Dylan’s contributions lie in: 1. Providing structured, interpretable reward shaping to accelerate learning in sparse-reward settings. 2. Enabling option-level composition via differentiable symbolic reasoning, allowing agents to solve multi-step tasks that are otherwise intractable for standard RL approaches.
>
> Since existing methods do not address logic option composition or provide reward shaping within a differentiable planner, we did not include them as direct baselines. That said, we fully acknowledge the value of expanding the baseline set, and if the reviewer has specific suggestions, we would gladly incorporate them in the revision.
>
> [1] Guo J, Zhang R, Peng S, et al. Efficient symbolic policy learning with differentiable symbolic expression[J]. Advances in neural information processing systems, 2023.
>
> [2]Jiang Z, Luo S. Neural logic reinforcement learning[C]//International conference on machine learning.  2019.
>
> [3] Shindo H, Delfosse Q, Dhami D S, et al. BlendRL: A Framework for Merging Symbolic and Neural Policy Learning[C]//The Thirteenth International Conference on Learning Representations. 2025

---

> > ### Author Rebuttal · Reviewer_8u2F · 2026-04-02
> >
> > I appreciate the clarifications, but I'm still unable to make much sense of the "planner" in the "differentiable planner" proposed. The ability to generate plans depth-first or breadth-first does not seem to be enough, as neither type of search scales up to large spaces. At the same time, aren't systems like Alpha/Mu-Zero, Dreamer, and other model-based RL methods, also using (informed) different planning, except that the search us guided by learned heuristics and values? It looks to me that the key feature of the approach is the ability to combine learned control and symbolic, "background" knowledge -- I don't see the "differential planning" part as clearly, as original, or as important, when viewed in the context of deep, model-based RL. On the one hand, the meta rules can capture non-greedy exploration strategies like BFS, but the greedy approaches also do planning, informed by the learned values and policies.

---

> > > ### Author Response · Authors · 2026-04-06
> > >
> > > We thank the reviewer for their continued engagement. Our submission is driven by a fundamental research question: *how can common knowledge, (including but not limited to knowledge implicitly encoded in LLMs), be leveraged to improve the efficiency of reinforcement learning?* Current RL practice largely ignores such knowledge, forcing agents to rediscover simple concepts like “keys open doors” through trial and error. This is unnecessarily wasteful. Our contribution is an interpretable and verifiable symbolic planner that serves as a differentiable interface between common knowledge and the RL agent. In particular, the planner is used to encode domain knowledge in a form that can directly guide and accelerate policy learning. Moreover, in our framework, the planner is not meant to replace the policy; rather, it encodes structured domain knowledge that helps guide the policy toward more sample-effective learning. We believe this differentiability is the key distinction from prior symbolic approaches and is crucial for automated, task-specific knowledge extraction. **This is fundamentally different from model-based RL methods** such as AlphaZero, MuZero, and Dreamer. In model-based RL, planning is typically performed in a learned latent or predictive model of the environment: the agent simulates future trajectories, evaluates them using learned value functions or policies, and selects actions based on these predicted outcomes. As also noted in Lines 119–126 of our submission

---

### Official Review · Reviewer_y4YG · 2026-03-13

**Soundness:** 2
**Presentation:** 2
**Significance:** 2
**Originality:** 3
**Overall Recommendation:** 3
**Confidence:** 3

**Summary:**

This paper proposes differentiable symbolic planner (**DYLAN**). DYLAN is a framework that integrates symbolic planning into deep reinforcement learning to address sparse rewards and generalization bottlenecks. DYLAN has two primary roles: as a reward model, it provides structured intermediate feedback by decomposing complex goals into logical subtasks; and as a planner, it composes logic options to enable zero-shot generalization to novel tasks. By parameterizing logic rules as tensors and optimizing rule weights via gradient descent, DYLAN adaptively selects search strategies (e.g., BFS vs. DFS) and achieves significantly faster convergence compared to traditional RL methods.

**Compliance With Llm Reviewing Policy:**

Affirmed.

**Final Justification:**

I will maintain my negative score as concerns raised by me and other reviewers cannot be addressed with minor efforts. I encourage authors conduct more experiments and analysis and submit a top AI venue in the next cycle.

**Key Questions For Authors:**

- Your framework requires a "manual" or predefined logic rules to initialize the planner. How can this approach be adapted for autonomous discovery in environments where such prior knowledge does not exist?
- Given that real-world signals are rarely as clean or well-defined as the grid-world symbolic states used in your experiments, how does DYLAN perform when the abstraction function $\phi$ is noisy or produces incorrect symbolic interpretations?
- Could you provide experimental results in more complex and higher-dimensional environments to further demonstrate the scalability and practicality of the proposed method？

**Limitations:**

Yes

**Strengths And Weaknesses:**

**Strengths**
- The framework effectively mitigates the exploration bottleneck in sparse-reward environments by providing structured, human-aligned reward shaping, leading to substantially faster convergence compared to standard deep RL baselines.
- By composing logic options, DYLAN enables agents to solve novel, multi-step tasks at test time without any additional training or goal-specific fine-tuning, demonstrating compositional flexibility over end-to-end neural networks approaches.
- As a differentiable symbolic planner, DYLAN successfully bridges logical reasoning with gradient-based optimization, allowing it to adaptively learn rule weights and dynamically adjust search strategies to suit specific task demands. This is novel.

**Weaknesses**
- The primary weakness is the framework's heavy reliance on perfectly defined "toy" environments. In real-world applications, logic states (e.g., "key pressed") are rarely provided as clean, noise-free signals; they must be inferred from messy, high-dimensional sensor data. The paper fails to demonstrate how DYLAN would handle the ambiguity and perception errors inherent in non-gridworld and high-dimensional settings.
- Although the authors use LLMs to assist in rule generation, the pipeline still heavily depends on human experts to verify and "fix" the logic rules before they can be used for training.

---

> ### Author Rebuttal · Authors · 2026-03-30
>
> We thank the reviewer for highlighting the paper’s strengths on sparse-reward exploration, compositional flexibility, and the differentiable planner. Here are the detailed replies to the questions.
>
> **Q1** and **Q2:** While our current setup assumes access to symbolic states and rules, we regard symbol acquisition and rule extraction as important but separate lines of work that fall outside the scope of this paper. A potential mitigation strategy is to leverage stronger VLMs that can reliably provide robust, structured knowledge, in line with recent efforts to improve symbolic domain learning robustness [1] or to extract domain language directly from LLM [2][3][4]. Another promising direction is the use of unsupervised symbolic abstraction techniques for automatically extracting symbolic rules or concepts, such as those proposed in [5][6].
> Please note that our work focuses on how to leverage domain knowledge to guide reinforcement learning in a differentiable manner. To the best of our knowledge, there is no existing method that can achieve this. Moreover, we view perfect rule and state induction systems as an important but separate line of work that falls outside the scope of this paper.
>
> [1]Claudius Kienle, Benjamin Alt, Oleg Arenz, and Jan Peters. Lodge: Joint hierarchical task planning and learning of domain models with grounded execution. arXiv preprint arXiv:2505.13497, 2025.
>
> [2] Silver T, Dan S, Srinivas K, et al. Generalized planning in pddl domains with pretrained large language models[C]//Proceedings of the AAAI conference on artificial intelligence. 2024, 38(18): 20256-20264.
>
> [3] Pallagani V, Muppasani B C, Roy K, et al. On the prospects of incorporating large language models (llms) in automated planning and scheduling (aps)[C]//Proceedings of the International Conference on Automated Planning and Scheduling. 2024, 34: 432-444.
>
> [4] Liang J, Huang W, Xia F, et al. Code as policies: Language model programs for embodied control[C]//2023 IEEE International conference on robotics and automation (ICRA). IEEE, 2023: 9493-9500.
>
> [5]Jingyuan Sha, et al. Neuro-symbolic predicate invention: Learning relational concepts from visual scenes. Neurosymbolic Artificial Intelligence, 1:NAI–240712, 2025
>
> [6] Wolfgang Stammer, et al.. Neural concept binder. Advances in Neural Information Processing Systems, 37:71792–71830, 2024
>
> **Q3:** The environments used in our evaluation, including MiniGrid, Atari, and BabyAI, are high-dimensional and challenging. For example, as illustrated in Figure 3, PPO fails to converge on the image-based MiniGrid DoorKey 16×16 task. These environments are recognized as challenging, as also noted in [7, 8]. Furthermore, we include a quadruped navigation task (Lines 312–317 and in App M), which shows that Dylan, as a reward model, also improves reinforcement learning in continuous action spaces.
>
> While our high-level planner is limited to tractable symbolic domains, the underlying MDP can still be high-dimensional and continuous, as demonstrated by our vision-based MiniGrid experiments and the quadruped task. We will revise the paper to make this point clearer.
>
> [7] Roger Creus Castanyer and Faisal Mohamed and Pablo Samuel Castro and Cyrus Neary and Glen Berseth. ARM-FM: Automated Reward Machines via Foundation Models for Compositional Reinforcement Learning,  International Conference on Learning Representations, 2026
>
> [8] Tang H, Key D, Ellis K. Worldcoder, a model-based llm agent: Building world models by writing code and interacting with the environment[J]. Advances in Neural Information Processing Systems, 2024, 37: 70148-70212.

---

> > ### Author Rebuttal · Reviewer_y4YG · 2026-04-04
> >
> > Thanks for your response and the introduction of related works [1-6] and [7-8]. However, I do not think my concerns could be addressed solely by listing papers. I will keep my score.

---

> > > ### Author Response · Authors · 2026-04-08
> > >
> > > We thank the reviewer for their continued engagement. We believe these concerns point to an important future work of the framework, rather than to the validity of the paper’s core contribution.
> > >
> > > Regarding noisy sensory inputs, we would like to point to RQ2, where incomplete visibility of the orinigal MDP does not prevent Dylan from improving the agent’s performance. We also want to clarify that the current experimental evidence is not limited to “toy tasks.” we evaluate Dylan on image-based MiniGrid and Atari settings. Moreover, we include a quadruped navigation task (App. M), which involves a more challenging environment with continuous action spaces, this simulation was commonly shown to transfer to the real robot [1,2,3]. In addition, our imperfect-knowledge experiments (App. L) test robustness when the available symbolic knowledge is incomplete. We agree with the reviewer that the current submission does not yet address noisy symbol abstractions or end-to-end rule extraction. However, we view these two aspects as important future directions that fall outside the scope of current submission .
> > >
> > > Our submission is driven by the following research question: *how can common knowledge (including but not limited to knowledge implicitly encoded in LLMs), be leveraged to improve the efficiency of reinforcement learning?* Current RL practice largely ignores such prior knowledge, forcing agents to rediscover simple concepts such as “keys open doors” purely through trial and error. We argue that this is unnecessarily inefficient. Our contribution is therefore an interpretable and verifiable symbolic planner that serves as a differentiable interface between common knowledge and the RL agent.  We will revise the manuscript to make this intended scope much more explicit.
> > >
> > >
> > > [1] Zakka K, Tabanpour B, Liao Q, et al. Mujoco playground[J]. arXiv preprint arXiv:2502.08844, 2025.
> > >
> > > [2] Luo J Y, Song Y, Klemm V, et al. Residual policy learning for perceptive quadruped control using differentiable simulation[C]//2025 IEEE International Conference on Robotics and Automation (ICRA). IEEE, 2025: 1-8.
> > >
> > > [3] Alvarez-Padilla J, Zhang J Z, Kwok S, et al. Real-time whole-body control of legged robots with model-predictive path integral control[C]//2025 IEEE International Conference on Robotics and Automation (ICRA). IEEE, 2025: 14721-14727.

---

### Official Review · Reviewer_Fgn3 · 2026-03-14

**Soundness:** 3
**Presentation:** 2
**Significance:** 3
**Originality:** 2
**Overall Recommendation:** 3
**Confidence:** 4

**Summary:**

The paper proposes a framework that uses LLLM-generated logical primitives together with differentiable symbolic inference to help reinforcement learning. As I understand it, the basic idea is to use prior knowledge to produce a more structured reward (pseudo/intrinsic rewards) or planning signal over intermediate subtasks, so that the agent does not have to discover everything from scratch through sparse reward alone. The paper presents this both as a differentiable reward model and as a way of composing higher-level behaviors, and evaluates the approach on exploration and generalization benchmarks.

**Compliance With Llm Reviewing Policy:**

Affirmed.

**Key Questions For Authors:**

See main comments for more context, just repeating those questions I already brought up in my review above:

1. The paper claims to be the first differentiable symbolic planner: in what sense? are you sure sure about that ?

2. How should the method be understood from a state representation point of view? If plan execution requires tracking progress, is the intended view an augmented-state MDP, or something closer to a history-based or partially observable formulation?

3. Which parts of the reasoning machinery are inherited from prior differentiable forward-reasoning work, and which parts are the actual contribution of Dylan? I found that hard to disentangle.

4. Why is logic the right prior here, as opposed to other ways of injecting structured intermediate guidance? I think stronger ablations, or at least a sharper discussion, would help.

**Limitations:**

Good job for documenting the prompting process in the appendix, this is very useful and transparent.

But again as explained in the main review, the specific benchmarks are not for me in any way a convincing demonstration that this is something I would need to use anywhere else other than in writing more papers on those same benchmarks.

**Strengths And Weaknesses:**

I think there is an interesting idea in here. The part I find actually interesting is not the generic "humans use priors" story, but that you leverage LLM to generate logic primitives, and these are then combined with a differentiable inference or planning module to produce a reward signal. Very much in line with Motif, Mastero Motif, Eureka, but with logic instead. Because it uses logic instead of preferences or code, this is a meaningful addition to the broader family of methods that try to inject prior structure into RL via LLMs.

That said, I think the paper is not yet presenting itself in the strongest way.

First, the abstarct did not really work for me. The "humans solve tasks by decomposing them into subtasks" framing, the coffee example, and related language all felt very generic. I know this is common, and I am guilty of using that kind of framing too (the coffee example in particular), but at this point it reads like something we have all seen many times. Also, maybe this is partly a matter of taste, but "humans do X" is not, to me, a strong reason to build a method that does X. The practical question is whether this gives us a useful way to solve real problems better. I think the stronger motivation is simply that sparse-reward RL benefits from good intermediate structure, and that this paper proposes one particular way of providing that structure. I would encourage the authors to get to that point much faster in the abstract and intro.

On originality, I would be careful with the claim that this is "the first differentiable symbolic planner." "The first" is a strong claim, and those are increasingly hard to make safely with the volume of papers we see in our field. Maybe the authors are right under a narrow definition, but if so that definition needs to be made much sharper. As written, I found the claim too strong, especially given nearby recent work in differentiable symbolic RL. For example: https://neurips.cc/virtual/2023/poster/69994 ok this one doesn't use the same strips-like foundations, but still. And on top of that, most of the paper is as far as I understand, an application of Shindo et al., 2023 to RL. This is much more to me the right paper to claim that they are the first differentiable planner out there; not this paper.

A more conceptual issue is that I am a bit confused by the way the paper frames all of this under MDPs. This kind of approach seems to need some notion of memory to track where one is in the execution of a plan. In that sense, just presenting it under the umbrella of MDPs feels odd unless the paper explicitly explains the augmented-state construction. Otherwise the more natural view, at least to me, is history-based, or something closer to partial observability. I think the paper should say more about this. You cite the work of Sheila Mcilraith on reward machines, which is the right thing to do, but know see how this work spends of lot of time showing how they build this state machine to make sure that everything is amenable to an MDP? I think you need that too as well here.

Presentation-wise, the paper also becomes abrupt right when it reaches the technical content. When it starts introducing the planner formally, with valuations (v^{(0)}) and (v^{(t)}), it moves too quickly. Readers coming from RL are not necessarily comfortable with logic planning ideas, and I think this part needs more of a bridge. Relatedly, I was sometimes unsure what exactly is inherited from earlier differentiable forward-reasoning work and what is actually new here. At points the wording sounds like "we encode each planning rule," but elsewhere it becomes clear that some of the machinery is really coming from prior work. "we" is really Shindo. Perhaps that's you I don't know, but in this double-blind mode I can only assume that "we" is a misattribution.

I also think the differentiable inference piece is very important and needs to be highlighted much more in the main text. Right now I had to look in Appendix C to really understand what was going on there. Before that, I was still unsure whether this was relying on some search procedure or some black-box symbolic planner under the hood. If differentiable inference is the main mechanism, then that should be front and center, not something the reader has to figure out later.

More generally, I think the paper needs a clearer scientific hypothesis. There is a general temptation here, and again I am guilty of this too, to build something because it is possible to build, because it combines appealing ingredients, because the construction is neat. But as scientists, we really can't avoid our duty of being clear about what hypothesis is being tested. One reasonable version would be that good reward structure matters, and that if we can provide a sufficiently good prior in this logic-based form, learning improves. But then the next question comes naturally: why logic in particular? Why is this the right prior rather than some other structured reward prior, programmatic representation, or language-conditioned shaping mechanism? I do not think the current paper really answers that. This is where I think the ablations are still not doing enough.

And this connects to my broader concern about significance. MiniGrid, BabyAI, Atari, yes, these domains can be useful for isolating mechanisms, and again I am guilty of using them too. But I would really like the paper to push itself harder here. What actual class of concrete problems is this supposed to help with? Even if the experiments remain in benchmark land and that "everybody uses them therefore I have too", I think the discussion (at the bare minimum) should try harder to connect the method to real use cases rather than stopping at the level of "it helps on these standard environments."

Minor note: There are also a few places where the terminology needs more care. "Logic options" for example on line 70 in the intro. Clearly you mean the options framework here, but you can't just namedrop this without some kind of quotes, italics + citation. Otherwise it reads like a new term being introduced a bit too casually.

---

> ### Author Rebuttal · Authors · 2026-03-30
>
> We appreciate the reviewer’s careful reading, thoughtful feedback, and positive assessment of the central idea. We are grateful for the time and effort the reviewer invested in helping improve the paper. Below, we provide detailed responses to the questions.
>
> **Q1** and **Q3:**  To the best of our knowledge, we are confident that this is the first differentiable symbolic planner, and especially  the first work to use a differentiable planner for reward shaping in RL. While there are prior works on symbolic planning in reinforcement learning ([1],[2]) and on differentiable reasoning ([4],[5],[6]), our method differs from these works.
>
> Symbolic planners, such as those used in [1][2] or PDDL [3], are inherently non-differentiable. In contrast, differentiable reasoning methods such as [3][4][5] are differentiable but typically lack planning capability, which requires generating action sequences. Our work bridges this gap by extending differentiable reasoning with explicit planning capability using meta reasoning, which we consider as a contribution in itself.
>
> Besides being the first differentiable symbolic planner, Dylan’s novelty lies in how differentiable planning is tightly integrated into the RL pipeline (1) provides structured reward shaping that accelerates learning and exploration for RL agents; and (2) enables differentiable symbolic option composition, allowing agents to solve multi-step tasks that are otherwise difficult for classical RL under sparse feedback and long horizons.
>
> **Q2:** We use augmented state representation for reward to guide the exploration, but only provide the original state representation to the policy, which enables us to solve the original MDP.
>
> **Q4:** We are convinced that complex learning tasks benefit from the use of common knowledge to guide learning. We fully agree that logic is not the only possible mechanism for incorporating such knowledge as intermediate guidance. Our choice of logic is motivated by three properties that are especially valuable in our setting: interpretability, verifiability, and differentiability.
>
> Moreover, when extended into a differentiable planning formulation, it can represent probabilities over alternative plans. This allows us to quantify how likely the agent, from its current state to eventually achieve the final goal, as demonstrated by our adaptive reward formulation.  Looking ahead, the overall framework remains end-to-end differentiable, also opens the door to jointly optimizing the planning component and the individual options, allowing plans to adapt as the options improve.
>
> [1] Kokel, Harsha, et al. "Reprel: Integrating relational planning and reinforcement learning for effective abstraction." Proceedings of the International Conference on Automated Planning and Scheduling. Vol. 31. 2021.
>
> [2] Lyu, Daoming, et al. "SDRL: interpretable and data-efficient deep reinforcement learning leveraging symbolic planning." Proceedings of the AAAI conference on artificial intelligence. Vol. 33. No. 01. 2019.
>
> [3] Silver T, Dan S, Srinivas K, et al. Generalized planning in pddl domains with pretrained large language models[C]//Proceedings of the AAAI conference on artificial intelligence. 2024, 38(18): 20256-20264.
>
> [4]Shindo H, Pfanschilling V, Dhami D S, et al. α ilp: thinking visual scenes as differentiable logic programs[J]. Machine Learning, 2023, 112(5): 1465-1497.
>
> [5]Evans R, Grefenstette E. Learning explanatory rules from noisy data[J]. Journal of Artificial Intelligence Research, 2018, 61: 1-64.
>
> [6]Manhaeve R, Dumancic S, Kimmig A, et al. Deepproblog: Neural probabilistic logic programming[J]. Advances in neural information processing systems, 2018, 31.

---

### Decision · Program_Chairs · 2026-04-30

**Decision:**

Reject

**Comment:**

This paper proposes a novel differentiable symbolic planner designed to enable reward shaping that incorporates human priors to shape subtask rewards. The reviewers generally found the approach to be a compelling one, but there were a wide variety of concerns that left their opinions borderline. Most notably, the experiments are largely based on toy benchmarks that do not exhibit real-world complexities (e.g., partial observability, noisy/inconsistent symbols, etc.) that have traditionally posed a significant challenge for symbolic approaches. Showing evidence that the proposed approach can bridge these gaps would significantly strengthen the paper. The authors point to two experiments to try and address these claims: (1) Atari; however, Atari is now considered to be a toy environment due to the lack of noise and the limited impact of partial observability, and (2) a quadruped simulation; however, this experiment is very minimally described in the paper, and quadrupeds are to some extent considered a solved problem. I would recommend considering more complex environments such as robot arms and object manipulation. More broadly, there were also concerns about the quality of the writing and the comparison to existing work on differentiable planning; addressing these concerns would also strengthen the paper.